# CO₂ emissions of drained coastal peatlands in the Netherlands and potential emission reduction by water infiltration systems

Ralf C.H. Aben[1], Daniël van de Craats[2], Jim Boonman[3], Stijn H. Peeters[1], Bart Vriend[3], Coline C.F. Boonman[1], Ype van der Velde[3], Gilles Erkens[4,5], Merit van den Berg[3]

[1]Department of Ecology, Radboud Institute for Biological and Environmental Sciences, Radboud University, Nijmegen, 6525 AJ, the Netherlands
[2]Soil, Water and Land use, Wageningen Environmental Research, Wageningen, 6708 PB, the Netherlands
[3]Faculty of Science, Department of Earth Sciences, Vrije Universiteit Amsterdam, Amsterdam, 1081 HV, the Netherlands
[4]Deltares Research Institute, Utrecht, 3584 BK, the Netherlands
[5]Department of Physical Geography, Utrecht University, Utrecht, 3584 CS, the Netherlands

*Correspondence to*: Ralf Aben (Ralf.Aben@ru.nl)

**Abstract.** Worldwide, drainage of peatlands has turned these systems from $CO_2$ sinks into sources. In the Netherlands, where ~7 % of the land surface consists of peatlands, drained peat soils contribute >90 % and ~3 % to the country's soil-derived and total $CO_2$ emission, respectively. Hence, the Dutch Climate Agreement set targets to cut these emissions. One potential mitigation measure is the application of subsurface water infiltration systems (WIS) consisting of subsurface pipes connected to ditch water. WIS aims to raise the water table depth (WTD) in dry periods to limit peat oxidation while maintaining current land-use practices. Here, we used automated transparent chambers in 12 peat pasture plots across the Netherlands to measure $CO_2$ fluxes at high frequency and assess 1) the relationship between WTD and $CO_2$ emissions for Dutch peatlands and 2) the effectiveness of WIS to mitigate emissions. Net ecosystem carbon balances (NECB) (up to four years per site, 2020–2023) averaged 3.77 and 2.66 t $CO_2$-C ha$^{-1}$ yr$^{-1}$ for control and WIS sites, respectively. The magnitude of NECBs and slope of the WTD-NECB relationship fall within the range of observations of earlier studies in Europe, though they were notably lower than those based on campaign-wise, closed chamber measurements. The relationship between annual exposed carbon (defined as total amount of carbon within the soil above the average annual WTD) and NECB explained more variance than the WTD-NECB relationship. The magnitude of the NECB represented 1.0 % of the annual exposed C on average, with a maximum of 2.4 %. We found strong evidence for a reducing effect of WIS on $CO_2$ emissions—reducing emissions by 2.1 (95% confidence interval 1.2–3.0) t $CO_2$-C ha$^{-1}$ yr$^{-1}$—and no evidence for an effect of WIS on the WTD-NECB and annual exposed carbon-NECB relationships. This means that relationships between either WTD or exposed carbon and NECB can be used to estimate the emission reduction for a given WIS-induced increase in WTD or exposed carbon. High year-to-year variation in NECBs calls for multi-year measurements and sufficient representative measurement years per site as demonstrated in this study with 35 site-years observations.

# 1 Introduction

Peatlands only cover 3 % of the Earth's surface, yet they store 30 % of global soil carbon (C), and thereby function as an important global C sink (Friedlingstein et al., 2022; Leifeld & Menichetti, 2018; Yu et al., 2010). Peatlands consist of non- or partly decomposed plant material and are typically formed under wet and anoxic conditions when supply of dead plant material exceeds decomposition. However, many peatlands worldwide have been drained and claimed for human purposes—mainly agriculture and forestry—during the last centuries (Kaat & Joosten, 2009; UNEP, 2022). Drainage immediately halts peat formation and increases soil aeration, which in turn accelerates aerobic microbial peat decomposition. This effectively reverses a peatland's function as a $CO_2$ sink by emitting large amounts of $CO_2$—sequestered over thousands of years—back into the atmosphere (Erkens et al., 2016; Evans et al., 2021; Tiemeyer et al., 2020). Worldwide, drained peatlands are responsible for 2–5 % of the total anthropogenic greenhouse gas (GHG) emission (Bonn et al., 2016; Humpenöder et al., 2020; Leifeld & Menichetti, 2018). Given the high $CO_2$ emissions from drained peatlands, reducing this emission would be a prerequisite to reach targets set by the Paris Climate Agreement to keep global warming below 1.5–2.0 °C (Leifeld & Menichetti, 2018). Hence, prompt measures are needed to limit $CO_2$ emissions from peatlands.

The Netherlands arguably has the longest history of intensive drainage and exploitation of peat soils in the world (Erkens et al., 2016). Currently, about 290.000 ha (ca. 7 % of the Dutch land surface) consists of peat soils of which ca. 77 % is used for agriculture, primarily as pastures for dairy farming (Arets et al., 2021). Due to deltaic and coastal conditions 17 % and 36 % of coastal peatlands in the Netherlands are covered by a thick (40–80 cm) and thin (<40 cm) clay cover, respectively (Jansen et al., 2009). Cultivated, drained peat soils in the Netherlands emit an estimated 4 Mt $CO_2$ per year (Arets et al., 2021), constituting ca. 3 % of the country's total $CO_2$ emission (CBS, 2023). The Dutch Climate Agreement (Ministry of Economic Affairs and Climate Policy, 2019) targets a reduction of 1 Mt $CO_2$-eq. per year from drained peat areas by 2030 and a 95 % reduction of emissions by 2050 relative to 1990. Hence, there is an urgency to explore, test and apply emission mitigation measures in drained peatlands.

Most proposed mitigation measures rely on limiting or reversing drainage of peatlands, thereby (temporarily) decreasing water table depths (WTD). A shallow WTD decreases the extent of the unsaturated zone, limiting the maximum depth of oxygen intrusion into the soil (Boonman et al., 2024b), thereby mitigating aerobic decomposition and $CO_2$ emissions. There are indeed several studies that show a clear relationship between WTD and $CO_2$ emission, although they differ in the type of relation and magnitude of emissions. Some studies suggest a linear relationship between WTD and $CO_2$ emission (e.g. Couwenberg et al., 2011; Evans et al., 2021). Others, such as Tiemeyer et al. (2020) and Koch et al. (2023) found a relationship that fitted best with a sigmoid function, whereby changes in WTD at depths beyond 30 cm hardly affect $CO_2$ emission (meaning that raising the WTD is only useful at shallow depths). Of these studies, $CO_2$ emissions reported in

Tiemeyer et al. (2020) were the highest, being a factor 1.7 and 7.4 higher for a WTD between 0.2–0.4 m compared to Couwenberg et al. (2011) and Evans et al. (2021), respectively.

Several land management strategies are available to decrease peatland drainage, peat decomposition and the corresponding $CO_2$ emissions. Options include complete peatland rewetting for nature restoration (Nugent et al., 2019) or paludiculture (Abel & Kallweit, 2022; Wichtmann & Joosten, 2007), which are effective to limit peat oxidation (Tanneberger et al. 2022; Buzacott et al., 2023; van den Berg et al., 2024), but also means moving away from conventional agricultural land use. To maintain conventional agricultural use, alternative options include raising ditch water levels or applying (sub)surface water infiltration systems (WIS; e.g. Boonman et al., 2022; van den Akker et al., 2008; Weideveld et al., 2021) to reduce peat oxidation, albeit to a lesser extent than complete rewetting. In the Dutch coastal peatland areas, WIS consist of regularly spaced subsurface drains (commonly 4–6 m drain spacing), which are connected to ditches or to a managed reservoir. These systems allow for a more homogeneous WTD within a field, thereby decreasing the extent of the unsaturated zone in warm and dry summers. As spacing between ditches in these areas commonly is large (30–100 m), raising ditchwater levels would be less efficient than WIS in reducing the unsaturated zone thickness further away from the ditch, as the hydraulic conductivity of degraded peat soils is mostly low (Jansen et al., 2007; Kechavarzi et al., 2007; H. Liu et al., 2016). By reducing the unsaturated zone, application of WIS is expected to reduce aerobic peat decomposition and associated $CO_2$ emissions, while allowing conventional agricultural activities to continue. However, the effectiveness of WIS in terms of $CO_2$ emission reduction varies, since some studies found evidence for a decrease in yearly $CO_2$ emissions from WIS systems (Boonman et al., 2022; Boonman et al., 2024a; Offermanns et al., 2023; van den Akker et al., 2008), while other studies found insufficient evidence (Weideveld et al., 2021) or even found evidence for an increase in $CO_2$ emissions (Tiemeyer et al., 2024). Differences in reported effectiveness may be caused by differences in soil properties or hydrological boundary conditions (ditch water level, seepage, summer drought or wet conditions) among others.

This study presents the measurement results from a novel $CO_2$ emission monitoring network for Dutch coastal peatlands under intensive agricultural use using automated transparent chambers. The aim of this network is twofold: 1) to establish a relationship between WTD and annual $CO_2$ emissions, and its uncertainty, for this specific type of peatlands, and 2) to determine the effectivity of WIS as a measure to reduce $CO_2$ emissions from these peatlands. We derived annual net ecosystem carbon balance (NECB) estimates for six locations for up to four years (2020–2023) from high-frequency $CO_2$ flux measurements with automated transparent chambers. We then evaluated the relations between WTD and NECB estimates and determined the WIS effectiveness in terms of annual NECB differences in relation to effective changes in WTD.

## 2 Methods

Six locations distributed over the coastal peat areas in the Netherlands were selected (Sect. 2.1, Fig. 1, Table 1) for this study. The locations were instrumented with automated transparent chambers (Sect. 2.2) and environmental sensors (Sect. 2.3). Plots were harvested and fertilised (Sect. 2.4), resulting in several C input and export terms that are considered in the NECB estimates (Sect. 2.5).

### 2.1 Study sites and study setup

Six locations were selected where water infiltration systems (WIS; sometimes also called submerged drains) had (recently) been installed. These locations are distributed over the coastal peatlands (peatlands with surface level elevation below 1 m above mean sea level) in the Netherlands taking into account the following selection criteria: (1) the peat layer (>80 % organic matter) thickness exceeds 1 m, (2) is covered by less than 0.5 m of clay, and (3) locations are used as intensively managed grasslands which are mowed and/or grazed (Fig. 1). Five locations had both a control (CON; without WIS) and treatment field (with WIS); Table 1. One location (LAW) only consisted of a treatment field, and in one location (ZEG) we measured two different treatment fields which were compared with one control.

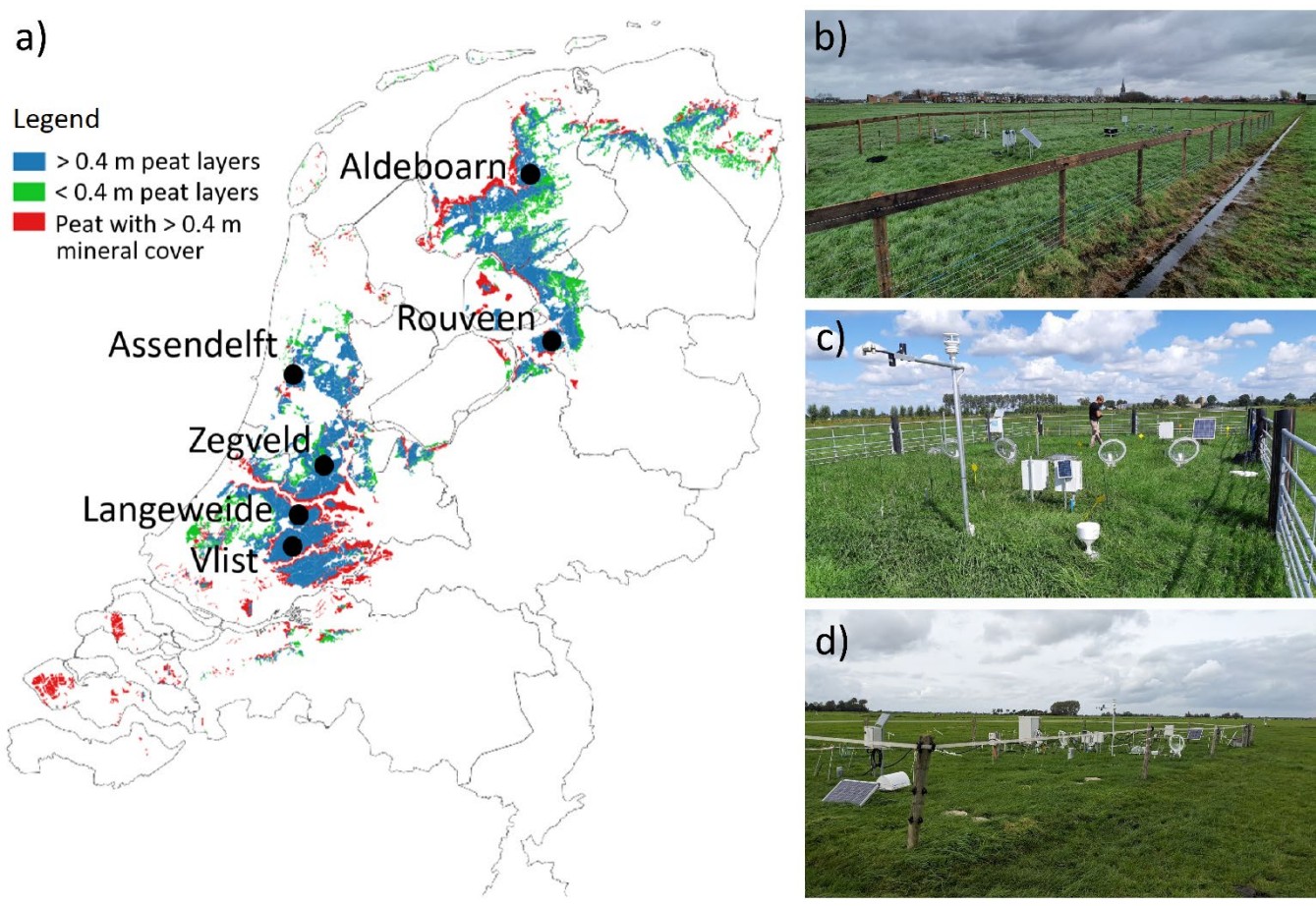

**Figure 1. Site locations and presence of coastal peatlands in the Netherlands (a). Photo impressions of Assendelft (b), Zegveld (c) and Aldeboarn (d).**

**Table 1. Overview of some characteristics of the measurement sites addressed in this paper, distinguishing the control (CON) and treatment (passive or active water infiltration; WIS) plots per location. Both listed WTD and ditch water table (WT) apply to**
115 **summer values. Peat thickness applies to the total Holocene peat layer thickness, clay thickness applies to the thickness of the clay(ey) layer on top of the peat. All units are in m.**

| Location | Treatment | WIS type | Chamber system | Targeted WTD | Targeted Ditch WT | Ditch spacing | Drain spacing | Year of drain install-ation | Peat thick-ness | Clay thick-ness |
|---|---|---|---|---|---|---|---|---|---|---|
| **Aldeboarn (ALB)** | CON | - | Eosense | - | 0.75–0.59$^x$ | 120 | - | - | 1.6 | 0.35 |
| | WIS | Passive | Eosense | - | 0.45$^x$ | 110 | 6.0 | 2016 | 1.7 | 0.4 |
| **Assendelft** | CON | - | VLUXpod | - | 0.45 | 185 | - | - | 2.0 | 0.3$^z$ |

| Location | System | Type | Device | | | | | Year | | |
|---|---|---|---|---|---|---|---|---|---|---|
| (ASD) | WIS | Active | VLUXpod | 0.25 | 0.45 | 185 | 4.0 | 2018 | 2.0 | 0.3[z] |
| Lange Weide (LAW) | WIS | Passive | VLUXpod-L | - | 0.4 | 62 | 6.0 | 2019 | 7.2 | 0.3 |
| Rouveen (ROV) | CON | - | Eosense | - | 0.4 | 36 | - | - | 3.1 | 0.3 |
| | WIS | Passive | Eosense | - | 0.4 | 42 | 8.0 | 2018 | 3.3 | 0.3 |
| Vlist (VLI) | CON | - | VLUXpod | - | 0.5 | 32 | - | - | >3.0[y] | 0.4 |
| | WIS | Passive | VLUXpod | - | 0.5 | 36 | 6.0 | 2011 | >3.0[y] | 0.4 |
| Zegveld (ZEG) | CON | - | Eosense | - | 0.55 | 65 | - | - | 6.8 | 0.3[z] |
| | WIS1 | Active | Eosense | 0.5 | 0.55 | 65 | 6.0 | 2016 | 6.8 | 0.3[z] |
| | WIS2 | Active | VLUXpod-L | 0.2 | 0.2 | 50 | 4.0 | 2020 | 6.5 | 0.3[z] |

[x] CON: Change in ditch WT from a ~ constant 0.75 m in 2021 to a fluctuating (range: 0.37–0.88 m) level thereafter. Range presented in table represents range in annual average ditch water table. WIS: Fluctuating ditch water level controlled by the farmer until March 2022, fixed at 0.45 m thereafter.

[y] Alternating layers of clay and peat. Total peat thickness exceeds 3 m.

[z] The top 0.3 m of the profile consists peaty clay or clayey peat.

The control fields were drained via ditches and, in some fields, furrows. Ditches always carried water and had a (more or less) fixed summer and winter water level, except for one location (ALB WIS, see Table 1). The treatment fields were drained via the same routes as the control fields, with the addition of a WIS. The WIS primarily increases water infiltration during dry (summer) periods, but also promotes drainage during wet (mostly winter) periods. Various configurations of WIS were used: subsurface drain tubes may be connected directly to the ditch below the water level (passive water infiltration system) or to a managed reservoir controlled by a pump (active water infiltration system). The latter system aims to actively maintain a target water table depth (WTD) in the field. The type of system per location is indicated in Table 1.

Measurements plots of the various locations in this study were set up in a similar fashion. A measurement plot of approximately 200 m$^2$ was fenced off. In the treatment plots, automated transparent flux chambers and subsurface sensors (Sect. 3.3) were installed in 3- or 4-fold (1) above or in proximity of a WIS-drain, (2) at a quarter distance between two WIS-drains and (3) midway between two WIS-drains (Fig. S1). In the control plot the same spatial distribution of measuring devices was used, although not related to the presence of a drain tube.

The soil C profiles across the study sites (Hefting et al., 2023) are visualized in Fig. 2. There is considerable variation in the average soil C content above the average WTD measured over the study period. In ALB this average soil C content was lowest (71–73 kg C m$^{-3}$), while it was highest in ZEG (122–148 kg C m$^{-3}$).

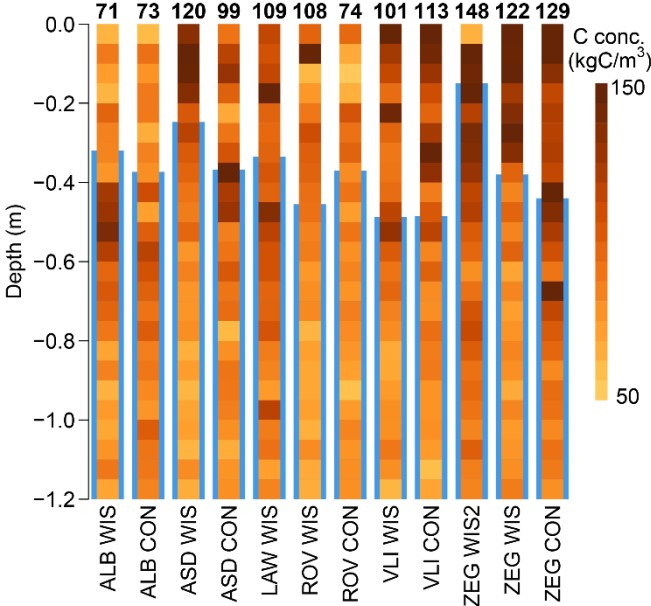

**Figure 2. Soil carbon profiles of all study sites (Hefting et al., 2023). The average water table depth (WTD) per plot is visualized by the blue bars in the background of each carbon profile, and the average carbon density (kg C m$^{-3}$) above the WTD is given above each profile.**

## 2.2 Automated transparent chamber CO$_2$ flux measurements

### 2.2.1 Chamber types

Fluxes of CO$_2$ between the soil-vegetation system and atmosphere were estimated from CO$_2$ concentration changes in closed chambers. For this, we used three types of automated transparent chamber systems (Table 1, Fig. 3). These automated chambers allow for continuous, day and night measurements of CO$_2$ concentrations at a high frequency. In all systems we used an infrared gas analyser (LI-850, LI-COR) to measure concentrations of CO$_2$ and H$_2$O that were logged by a Campbell CR1000x data logger once every two seconds.

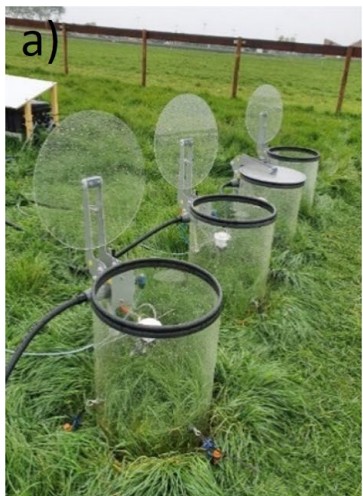 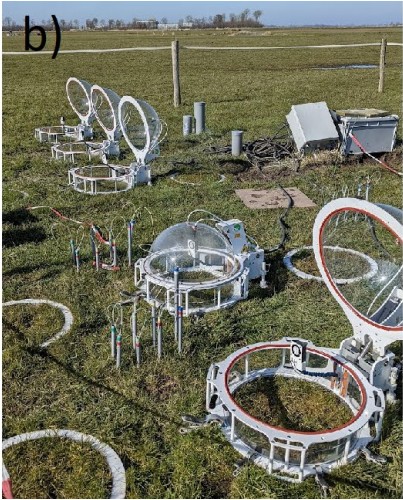 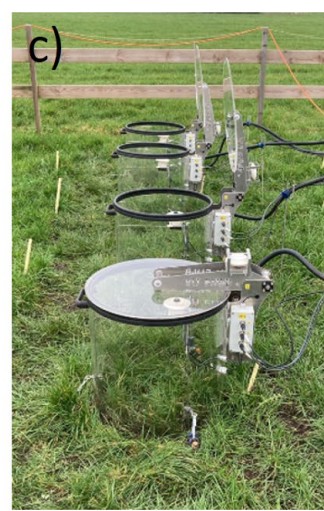

**Figure 3. Transparent chamber systems: (a) VLUXpod, (b) Eosense eosAC-LT and (c) VLUXpod-L chambers**

In ALB, ROV and ZEG (CON and WIS1), $CO_2$ fluxes on each plot were estimated using three eosAC-LT chambers (Eosense), connected to a multiplexer (eosMX; Eosense). Each chamber has a total height of 41.2 cm and volume of 72 L and consists of a transparent base (height: 15 cm; diameter: 52 cm) and transparent dome-shaped lid which was opened and closed by a linear actuator, closing in 15 to 30 seconds. Chambers were placed on permanent, serrated soil collars (15 cm deep). These collars offset the original chamber height by 0.5–6 cm, depending on soil swelling and shrinking; collar heights

for ALB and ROV were measured during site maintenance to adjust the volume used for flux calculations (see below). For ZEG CON and WIS, an average offset of 1 cm was used as no consistent measurements were available. All three chambers were connected to the multiplexer, which was used to control the chambers and route gas to the analyser. Recirculation of gas was achieved using the LI-850's built-in pump (0.75 lpm) and PTFE tubing to and from the chamber (8–10 m, one way). Every 30 minutes, chambers were measured sequentially with a 2.5-minute closure time and a 15–45 sec flushing period in

between.

    In ASD and VLI, a custom-built chamber system (referred to as 'VLUXpod' chambers) was used. Each system consisted of four transparent cylindrical chambers (volume ~62 L) with a base height of 50 cm, a diameter of 40 cm, and a transparent flat lid that was pneumatically controlled, which opened and closed within two seconds. In contrast to the Eosense chambers,

no permanent soil collar was used, but a custom-built tool was used to make 1–5 cm deep incisions into the soil to seal the chamber walls to the soil surface. Chamber height relative to the soil surface was measured when chambers were relocated. A multiplexer with an external pump (2.5 L min$^{-1}$; KNF NMP830KNDC-B 12V) was used to control the system and recirculate gas (8–10 m of polyurethane tubing, one way), from which gas was sampled by the analyser. Every 15 minutes chambers were measured sequentially using a 3-minute closure time and 15–45 seconds flushing in between.

A third system ('VLUXpod-L chambers') was used in ZEG WIS2 and LAW, which consisted of a similar setup as the aforementioned VLUXpod chambers. The main difference between the two was a larger diameter of 50 cm rather than 40 cm and the presence of a higher-flow gas circulation pump (5 L min$^{-1}$; KNF NMP830KPDC-B HP 12V).

### 2.2.2 Chamber operation

Chambers were moved and cleaned approximately every two weeks to limit lasting effects of chambers on conditions such as grass growth, soil temperature and soil moisture. The chambers were rotated over three rows (with three or four chambers per row, depending on the system), such that any chamber location was occupied approximately 33 % of the time. Grass heights were measured upon every chamber movement on all chamber rows. Chamber systems (including analysers) were removed from the field for maintenance and analyser calibration once every year. All chamber systems were equipped with a low-flow fan to achieve a well-mixed headspace (Christiansen et al., 2011; Rochette & Hutchinson, 2005).

### 2.2.3 Flux estimation

The $CO_2$ flux, hereafter named net ecosystem exchange (NEE, μmol $CO_2$ m$^{-2}$ s$^{-1}$), was calculated as

$$\text{NEE} = \frac{VP\left(1-\frac{W}{1000}\right)f}{RS(T+273.15)},\tag{1}$$

where $V$ (m$^3$) is the chamber volume, corrected for changes in collar or chamber height over time, $P$ (Pa) is the air pressure measured by each location's weather station, $W$ is the water vapor mole fraction as measured by the $CO_2$/$H_2O$ analyser (mmol mol$^{-1}$), $f$ is the rate of change in water-corrected $CO_2$ mole fraction (μmol mol$^{-1}$ s$^{-1}$) inside the closed chamber, $R$ is the ideal gas constant (8.314 Pa m$^3$ K$^{-1}$ mol$^{-1}$), $S$ (m$^2$) is the soil surface area and $T$ (°C) is the air temperature measured inside the chamber (VLUXpod chambers) or measured at 2 m height by the weather station (Eosense chambers). To determine $f$, we applied linear regression and a variety of regression periods. For each individual chamber system, regression periods were chosen such that only the linear portion of the concentration change was selected (Maier et al., 2022). This was required to limit effects of chamber closure that resulted in nonlinear concentration changes, such as (1) headspace $CO_2$ depletion and glass clouding during daytime and (2) spikes in $CO_2$ concentration that often occur immediately after chamber closure during nights with atmospheric stratification (Koskinen et al., 2014). As such, daytime regression lengths were restricted to a maximum of 30 to 60 seconds, starting just after the deadband (i.e., start of concentration change in response to chamber closure) to capture the initial slope, whereas night-time regression periods could be longer (up to 160 seconds) and started up to 100 seconds after chamber closure.

Data were left out from the flux calculation when analyser cell pressures or temperatures were outside of the calibrated operating range, gas concentrations were erroneous (e.g. due to IR-source failure) and in case of other types of system malfunctioning (e.g., non-functional fans or non-functioning chamber lids) or system maintenance. In some cases, a small

correction to the measured concentrations was applied based on drift in analyser calibration. A visual inspection of the data together with an automated quality control was applied to filter out other poor linear regression fits. The automated filtering procedure was based on a combination of regression fit characteristics, such as $r^2$, RMSE and actual flux slope. Thresholds for filtering deviated per chamber system and period considered.

## 2.2.4 Flux partitioning and gap-filling

For further processing we aggregated NEE fluxes by taking the mean of the measured fluxes of all chambers on a specific field over a half-hour period. Due to data quality control and system maintenance and malfunctioning, gaps were present in the aggregated flux data with extents ranging from half an hour to multiple weeks. We identified a gap as having no flux estimates from any of the chambers at the specific field during the half-hour period. An overview of the data availability per site is given in Fig. 4. To fill these gaps, as required to obtain an annual NECB estimate, we separated the 30-minute averaged net ecosystem exchange flux (NEE) into gross primary production (GPP) and ecosystem respiration ($R_{eco}$).

$$NEE = R_{eco} - GPP. \tag{2}$$

We model daytime $R_{eco}$ based on night-time $R_{eco}$, compensated for temperature differences only. Although it is common practice to model daytime $R_{eco}$ based on night-time $R_{eco}$ estimates, we acknowledge that it can lead to biased estimates due to divergent temperatures dependencies of day- and night-time $R_{eco}$ resulting from processes such as inhibited leaf respiration in light (Järveoja et al., 2020; Keenan et al., 2019).

$$R_{eco} = R_{ref} \cdot e^{E_0 \cdot \left( \frac{1}{(T_{ref} - T_0)} - \frac{1}{(T - T_0)} \right)}, \tag{3}$$

with $R_{ref}$ ($\mu$mol $CO_2$ m$^{-2}$ s$^{-1}$) is the reference respiration rate; $E_0$ (K) is the long-term ecosystem sensitivity coefficient to temperature; $T_{ref}$ (K) is the reference temperature for which the reference respiration was determined, $T_0$ (K) is the base temperature (set at 227.13 K, Lloyd and Taylor, 1994) and $T$ (K) is the observed soil temperature at 5 cm depth. To obtain a site-specific estimate of the long-term ecosystem sensitivity coefficient, Eq. (3) was applied to all measured, daily averaged night-time data for the whole timeseries at one location, with a reference temperature of 10 °C.

Daytime fluxes were partitioned based on the standard procedure as described by Falge et al. (2001), Oestmann et al. (2022), Tiemeyer et al. (2016) and Veenendaal et al. (2007). Given the site-specific value of E0, daytime $R_{eco}$ was modelled on a half-hourly basis using Eq. (3) with $R_{ref}$ and $T_{ref}$ given by the daily averaged night-time respiration rate and soil temperature at 5 cm depth, respectively, and T the measured soil temperature at 5 cm depth during the half-hour intervals. With the daytime calculated $R_{eco}$, an estimate of GPP was obtained using measured NEE and Eq. (2). GPP can be described by a rectangular hyperbolic light response curve (LRC) based on the Michaelis–Menten kinetic (Oestmann et al., 2022), given by

$$GPP = \frac{GPP_{2000} \times \alpha \times PAR}{GPP_{2000} + \alpha \times PAR - \frac{GPP_{2000}}{2000\mu mol\ m^{-2}s^{-1}} \times PAR},$$  (4)

where $GPP_{2000}$ ($\mu mol\ CO_2\ m^{-2}\ s^{-1}$) is the rate of C fixation at a PAR value of 2000; $\alpha$ ($\mu mol\ CO_2\ m^{-2}\ s^{-1}$ / ($\mu mol\ PAR\ m^{-2}\ s^{-1}$)) is the light use efficiency (the initial slope of the LRC) and PAR is the measured photosynthetically active radiation ($\mu mol\ m^{-2}\ s^{-1}$). As we determined GPP by partitioning, the (time-variant) parameters ($GPP_{2000}$ and $\alpha$) could be obtained on a daily base by fitting the LRC on the partitioned GPP.

In case of data gaps (Fig. 4) in the half-hourly aggregated data, $R_{eco}$ and GPP were gap-filled separately where daily obtained parameters from Eq. (3) and Eq. (4) (smoothened with a moving average of five days) were linearly interpolated. In the event of harvest, the moving average was cut off before and after harvest and LRC parameters were set to a minimum after harvest, to subsequently increase linearly to the obtained parameters five days after harvest. When a gap occurred over a harvest period, the parameters were taken up to three days before or after (in case of $R_{eco}$) harvest. If gaps were larger than this period, parameters were obtained from similar harvest moments from that site.

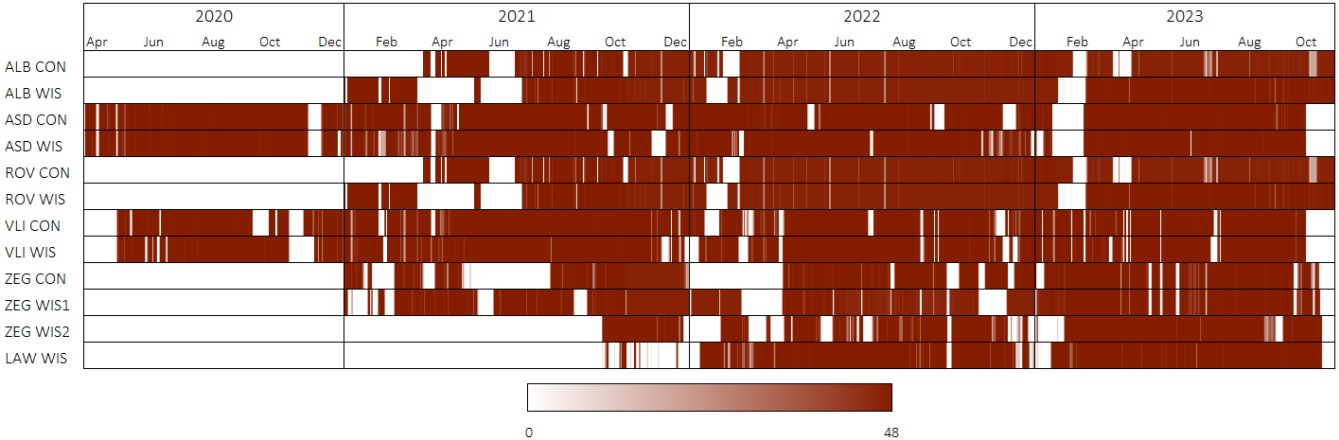

**Figure 4. Overview of CO$_2$ flux data availability on all sites. Red in different shades indicates data availability, where the darkest red refers to 48 half-hour data points per day and white to no data available. Sites were set-up at different moments; therefore, the starting dates of the flux measurements differ per site. Note that the periods depicted here as calendar years are not necessarily used to calculate annual net ecosystem carbon balances (Table S1).**

### 2.3 Environmental variables

On each of the plots we measured soil temperature (Drill & Drop probes, Sentek Technologies) and phreatic groundwater – and surface water levels (ElliTrack-D, Leiderdorp Instruments). Thirty-minute averaged soil temperatures were logged at 10 cm depth intervals from 5 to 115 cm depth. Phreatic groundwater levels were measured in monitoring wells, which were founded in deeper sand layers below the peat to assure a constant reference level. They were logged once every hour. Groundwater levels relative to the actual field height (i.e., WTD; Fig. S2) were calculated from surface movement measurements obtained from an extensometer (Van Asselen et al., 2020) combined with spirit levelling (four times a year) to

account for spatial differences in field height. Each of these variables were measured at least at three locations within each plot. For the WIS plot, these locations were next to the drain, at a quarter distance between drains and at midway between two drains. Meteorological measurements included air temperature and pressure (at a 30-minute logging interval), as measured in each location's control plot at 2.0 m height using a MaxiMet GMX500 (Gill instruments Limited). Precipitation was measured using an ARG314 tipping bucket rain gauge (Environmental Measurements Limited). PAR was measured at 1.8 m height (one minute logging interval) using a SKR 1840D (Skye Instruments).

We determined annual and summer mean WTD per plot ($WTD_a$ and $WTD_s$, respectively) by averaging the three measurement locations per plot, where annual refers to the total period of one year budget (Sect. 2.5), and summer refers to the months April up to September. The soil C profiles (Fig. 2) were used to determine the annual – and summer mean soil C exposure per plot ($Cexp_a$ and $Cexp_s$, respectively), taking the cumulative soil C amounts from soil surface to annual – and summer mean WTD.

**2.4 Harvest and fertilisation**

Plots were typically fertilised five times per year and mown five to nine times per year, aiming for at least once every four weeks during the growing season. Fertilisation was done with known quantities of mineral NPK (first two events) or N (remaining events) fertiliser for all sites, except for ALB, where manure was used as the latter is an organic farm. All sites used the same amounts and composition of mineral fertiliser (~250 kg N, 108 kg $P_2O_5$ and 195 kg $K_2O$ ha$^{-1}$ yr$^{-1}$). Applied manure and grass samples were weighed and analysed for C content.

To determine the C exported via grass harvests, grass yield was quantified for each chamber individually by weighing wet and dry (oven-dried for 48 h at 70 °C) biomass. For ALB and ROV, the average harvest per chamber per mowing event was determined as the average of grass yields collected from the different positions upon which the chamber is rotated, weighted by the amount of time that the chamber spent on each position. For other sites the grass was sampled only from the current chamber position. For ASD and VLI, differences in grass height on different chamber positions proved to be of minor importance. From the grass samples collected during each mowing event, the average and standard deviation (SD) of the C-export of the different chambers per harvesting event were calculated.

For ALB and ROV dried biomass samples were chopped using a cutting mill (SM 200, Retsch). Then, a homogenised subsample was ground using a mixer mill (MM 400, Retsch). Grounded biomass (4–5 mg) was weighed into tin capsules and analysed for C content using an NA 1500 elemental analyser (Carlo Erba). For all other locations, samples were sent to a commercial laboratory (Eurofins, Wageningen, the Netherlands) where they were thoroughly mixed and split into subsamples. The dried biomass was ground < 1 mm and C were determined using near-infrared spectroscopy (NIRS) performed on a Q-interline machine. The standard Eurofins Agro calibration curves for common Dutch grasslands (most

common species grown is *Lolium perenne*) were used, which are based on calibrations against wet chemistry (Harris et al., 2018).

## 2.5 Carbon budgets

The C budget of each site is given as the net ecosystem carbon balance (NECB) over a period of one year:

$$NECB = NEE + C_{\text{export}} - C_{\text{input}}, \tag{5}$$

with all terms are given in t C ha$^{-1}$ yr$^{-1}$ (Chapin et al., 2006). Positive C fluxes and budgets indicate a loss of C from the soil-vegetation system to the atmosphere. Note that the C budget in Eq. (5) does not account for C changes via runoff, lateral subsurface flow and emission of $CH_4$, CO and volatile organic C. Inorganic carbon is not added to the experimental fields and is also not widely present in the soil profile. Inorganic carbon is present in the soil water phase as dissolved $CO_2$ and $HCO_3^-$, mostly originating from peat decomposition. For the NECB calculation, we assume that changes in inorganic carbon in the soil profile including the water phase on a yearly basis (1-Jan–1-Jan) are small compared to the GPP and Reco fluxes and mostly fall within the uncertainty bounds of the overall fluxes. The input term in Eq. (5) consists of applied manure and is only relevant in ALB as other locations were fertilised with mineral fertiliser. The export term in Eq. (5) consists of harvested biomass, which is assumed to be released as $CO_2$ elsewhere during the year and factored in as loss from the system.

As measure of spatial heterogeneity, we also gap-filled half-hourly fluxes of each chamber individually and obtained the SD between the daily mean NEE fluxes in each chamber (SD$_{\text{NEE}}$). Further, if any day in the half-hourly chamber-averaged flux dataset consisted of less than 30 half-hour flux measurements, we added an extra gap-fill SD term (SD$_{\text{gap}}$), depending on the length of the gap. The term was determined by creating artificial gaps of 1, 5, 15 and 30 days, and comparing differences between measured data and gap-filled data. A linear relation was found between SD and gap size, which we extrapolated to obtain an estimate of the SD for any gap in the data. The SD of the fertilization C-import term (for ALB only) was estimated at 50 % of the total C import. The SD of the NECB resulting was then obtained with Eq. (6), as

$$SD_{\text{NECB}} = \sqrt{\sum SD_{\text{NEE}}^2 + \sum SD_{\text{gap}}^2 + \sum SD_{C_{\text{export}}}^2 + \sum SD_{C_{\text{import}}}^2}, \tag{6}$$

where each term is the sum of the occurrences in each year.

## 2.6 Statistics

All calculations and statistics were carried out in R (R Core Team, 2023). Pearson's correlation coefficient (r) was computed using the cor function of the 'stats' package. To establish relationships between NECB and potential predictors (i.e. WTD and Cexp), we used simple linear models (LM) using function lm of package 'stats'. To statistically compare the NECBs of the CON and WIS treatment we used linear mixed-effects models (LMM) using the lmer function of the 'lme4' package

(Bates et al., 2015a) with treatment as fixed effect. The effect of treatment on the relationship of NECB with potential predictors (i.e., WTD and Cexp), was tested using the interaction of treatment with the predictor of interest as fixed effects. To deal with the non-independence in the dataset (i.e. having multiple NECBs per location and per year) we treated measurement year nested in location as a random effect on the model's intercept for all LMMs mentioned above. In case the fitted LMM was evaluated to be (near) singular (i.e., NECB ~ $WTD_a$) due to the variance estimate of random effect 'year' being near zero, we ran the model with only location as a random effect, thereby following recommendations in Barr et al., (2013) and Bates et al. (2015b). To statistically compare the WTD-NECB relationship based on our data and those of other drained peatlands, we used NECB as the response variable, the interaction of WTD with the data source as fixed effect, and location as a random effect on the model intercept. We used type-III ANOVAs (function anova) to test the significance of the fixed effects of our various LMMs, with degrees of freedom and P-values calculated using the Kenward-Roger approximation (Kenward & Roger, 1997) integrated in the 'pbkrtest' and 'lmerTest' packages (Halekoh & Højsgaard, 2014; Kuznetsova et al., 2017). Model assumptions of linearity, homoscedasticity, and normality of residuals were checked using residual plots, histograms and Q-Q plots of residuals, and Shapiro-Wilk's test (function Shapiro.test of package 'stats'). When communicating our statistical results, we use the language of evidence as suggested by Muff et al., 2022.

## 3. Results and discussion

### 3.1 Chamber $CO_2$ flux estimates & carbon balances

We collected 12,485 daily $CO_2$ flux estimates, comprised of roughly 517,000 half-hourly means, based on ~3.1 million observed fluxes. We observed clear variability in the $CO_2$ fluxes for all locations and plots on temporal scales ranging from minutes to seasons (Fig. 5). The daily $CO_2$ fluxes ranged from -179 (net uptake) to 163 kg (net emission) of $CO_2$-C $ha^{-1}$ $d^{-1}$. The median daily $CO_2$ flux across all plots for the full study period was -5.1 kg $CO_2$-C $ha^{-1}$ $d^{-1}$. Highest daily net uptake rates were mostly confined to spring, while highest daily net emission rates generally occurred during summer. Aggregated half-hourly $CO_2$ flux data availability for the individual annual budget periods and sites considered was 83 % on average. We omitted the budgets of ALB WIS, ROV WIS and ZEG CON in 2021 from further analysis due to the low data availability and the large consecutive periods of missing data during the growing season (Fig. 4, Table S1) for which extensive gap filling was required.

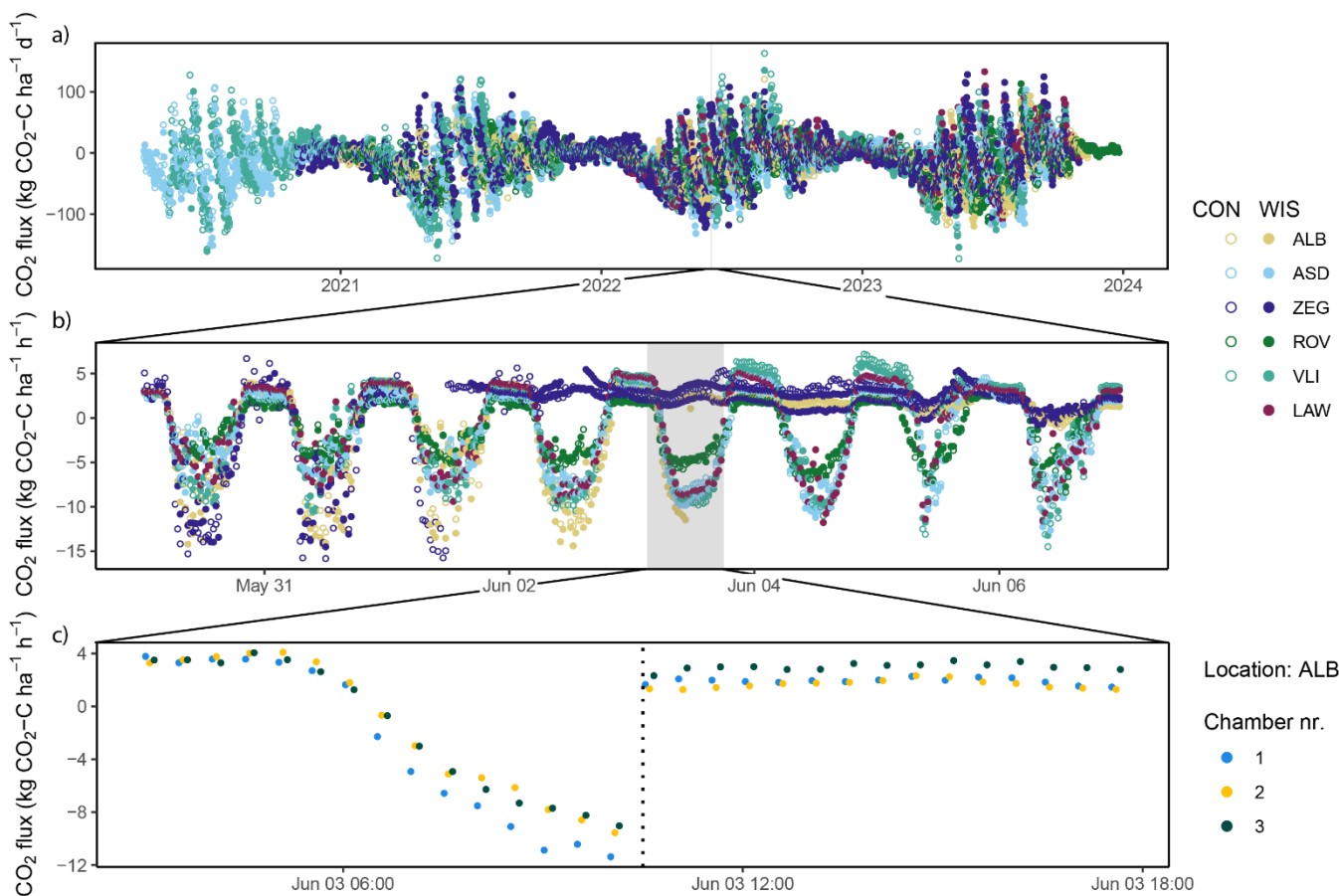

**Figure 5. Temporal variability of CO₂ fluxes observed at different timescales: (a) Daily means of each location and treatment (i.e., control [CON] and water infiltration system [WIS]) for the full study period; (b) Half-hourly means for each location and plot for the shaded period in (a); (c) CO₂ fluxes for each individual chamber of ALB CON for the shaded period in (b). The dotted line in (c) denotes a mowing event. Note that ZEG WIS has two timeseries included due to measurements at two different WIS sites.**

An overview of the annual C balances is presented in Fig. 6, distinguishing between NEE, harvest export and manure import. Harvest was especially high in 2020 and 2021. For most locations and years, it provides the largest C flux of the terms considered in this figure, with on average 6.4 t C ha$^{-1}$ yr$^{-1}$. This term is on the higher range of what was found on German peatland sites with *Lolium perenne* (1.3–6.4 t C ha$^{-1}$ yr$^{-1}$; Tiemeyer et al. (2020). Higher yields in our locations are likely due to high fertilization application and the frequent harvest events during growing season (~5–7 vs 1–5 cuts in Tiemeyer et al. [2020]). In ZEG WIS2, harvests are generally lower than in the WIS1 and CON plot of ZEG, likely owing to oxygen stress in the root zone (Bartholomeus et al., 2008) given the shallow WTD$_s$ (0.18, 0.49 and 0.64 m in WIS2, WIS1 and CON, respectively). The same applies for ASD WIS, where harvests are generally lower than for ASD CON (with an average WTD$_s$ of 0.30 and 0.55 m in WIS and CON, respectively). NEE terms are mostly negative and show quite some year-to-year

variation (Fig. 6). NEE was highest (i.e., close to zero) in LAW, VLI and ROV and lowest (i.e., strongest net uptake) in ALB and ASD.

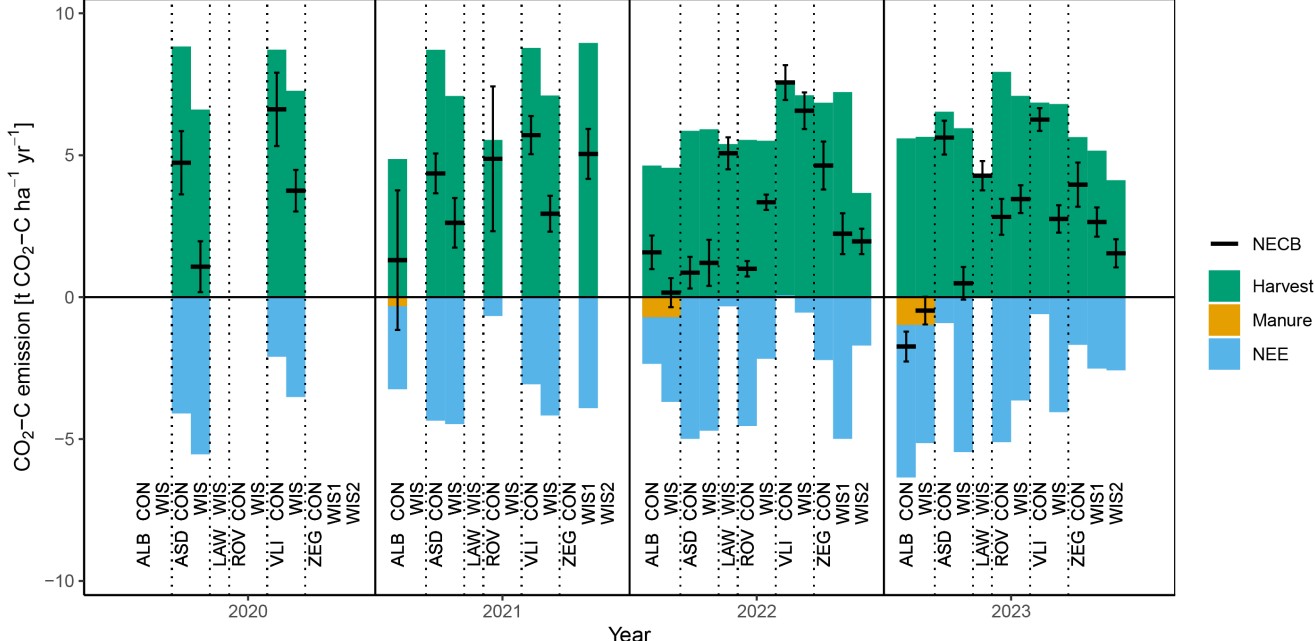

**Figure 6. Annual CO₂ emission terms (with uptake being negative) as net ecosystem exchange (NEE), harvest and manure (only in ALB) of the plots over the years 2020–2023. The black, horizontal lines indicate the net ecosystem carbon balance (NECB), including their standard deviations (indicated by whiskers). Specific values are also provided in Table S1.**

The estimated terms GPP and $R_{eco}$, being the two constituents of NEE, are substantially larger than the terms displayed in Fig. 6, with values ranging from -18 to -29 t C ha$^{-1}$ yr$^{-1}$ for GPP, and 14 to 25 t C ha$^{-1}$ yr$^{-1}$ for $R_{eco}$ (Table S1). The high harvest export term was also reflected in the GPP: in almost all cases, the uptake of C by plants (GPP) exceeded the respiration ($R_{eco}$), leading to negative NEEs (on average –3.0 t C ha$^{-1}$ yr$^{-1}$). On the contrary, in German peatland sites with

375 *Lolium perenne* the average NEE was +8.1 t C ha$^{-1}$ yr$^{-1}$ (Tiemeyer et al., 2020), while the average NEE of boreal and temperate peatlands used as grassland in Evans et al. (2021) was +1.3 t C ha$^{-1}$ yr$^{-1}$.

NECB, being the resultant of NEE, harvest export and manure import, shows a similar year-to-year variability as NEE and harvest export and averaged 3.17 t C ha$^{-1}$ yr$^{-1}$ across all site-years. In almost all cases the sum of NEE and harvest export led

to positive NECBs. Only in ALB we estimated NECBs to be ~0 in 2022 (WIS) and negative in 2023 (both WIS and CON). The substantial negative NECB estimates in 2023 are caused by a high negative NEE. This site, being the only site in the north of the Netherlands (Fig. 1), is quite different from the other sites in this study due to its 0.4 m thick clay cover, the lowest soil C content in the upper soil layer of all sites (Fig. 2), deviating peat composition, deviating land use history

(having experienced more deeply drained conditions) and ongoing manure application. Though these factors are likely to affect the magnitude of the NECBs, they cannot explain why the NECBs are negative, especially since positive NECBs (8.1–17.9 t C ha$^{-1}$ yr$^{-1}$) were found at the same site using campaign-wise measurements with manual chambers in 2017 and 2018 (Weideveld et al., 2021). In addition, NECBs of 2.8 and 6.4 t C ha$^{-1}$ yr$^{-1}$ were estimated for the ALB WIS and CON field, respectively, using eddy covariance measurements (period October 2021–October 2022; unpublished data). We currently do not have an explanation for the widely varying results and negative NECBs for this particular site, however these could be related to unquantified C fluxes, such as lateral transport of C via groundwater or C export via geese or mice or, a change in C storage in the root zone. Also, a dependency on the methods used to obtain the flux estimates (e.g. measurement technique and method of data processing) may be responsible for the varying results.

Another noticeable annual C budget is found in the CON plot of ASD in 2022. In this year we obtained an exceptionally low NECB in the CON plot compared to the other year budgets on that plot. In this specific year, chambers of the CON plot were moved to a different location within the plot, as the vegetation within the original chamber locations in this year was no longer representative for the vegetation within the plot. However, as there is no evidence of a high degree of spatial heterogeneity in e.g. soil parameters within the plot, we cannot appoint any concrete reasons why moving the chambers could have resulted in such a low NECB in the CON plot for this year.

**3.2 Relationships between NECB and controlling variables**

In Fig. 7 and Table 2 we show how the annual C budgets relate to WTD$_s$ (Fig. 7a, r$^2$ = 0.19) and WTD$_a$ (Fig. 7b, r$^2$ = 0.15) WTD. Because of warmer temperatures and deeper groundwater levels during summer, we expected the NECBs to relate substantially better to WTD$_s$ than to WTD$_a$ as proposed by Boonman et al. (2022). Our results, however, do not convincingly support this hypothesis and only show a slightly higher explained variance for the WTD$_s$. The similar performance of the two models is likely explained by the strong correlation between WTD$_a$ and WTD$_s$ (Pearson's r=0.88).

The relation between NECB and total exposed C within the soil profile above the average annual WTD (Cexp$_a$) was stronger (Fig. 7c, r$^2$ = 0.25) than the relation with WTD$_a$. The same is true for the relation between NECB and summer exposed C (Cexp$_s$) as compared to the relation with WTD$_s$ (r$^2$ of 0.26 and 0.19, respectively). Since relationships with Cexp explained more variance than those with WTD, we propose to use Cexp rather than WTD$_a$ as a predictor for NECB. The use of Cexp will be particularly important in the coastal zone and deltaic peatlands, because in these environments, flooding-derived clastic layers are commonly covering the peat layers (Koster et al. 2018).

**Table 2 – Linear model fits for NECB and explanatory variables related to water table depth (WTD) and exposed soil carbon (Cexp) presented in Fig. 7 and Table S1.**

| Explanatory variable | Function | $r^2$ | Figure |
|---|---|---|---|
| Summer water table depth | NECB = 6.53 $WTD_s$ + 0.03 | 0.19 | 7a |
| Annual water table depth | NECB = 8.45 $WTD_a$ + 0.05 | 0.15 | 7b |
| Summer exposed carbon | NECB = 0.0080 $Cexp_s$ + 0.15 | 0.26 | - |
| Annual exposed carbon | NECB = 0.0109 $Cexp_a$ – 0.06 | 0.25 | 7c |

The low and even negative NECBs from ALB, as mentioned in the previous section, are generally much lower than predicted by the linear regression and are positioned just inside or even outside the prediction intervals (Fig. 7a and b). However, when expressed against exposed soil C rather than WTD (Fig. 7c), these datapoints better approach predicted values, owing to the relatively low C stock in the upper part of these soils (Fig. 2). While Tiemeyer et al. (2016) showed that NECB relates to aerated soil N stock rather than C stock, our data suggests that exposed C does relate to NECB. We found that the magnitude of the NECB represented 1.1 % of the annual exposed C on average, with a maximum of 2.4 %.

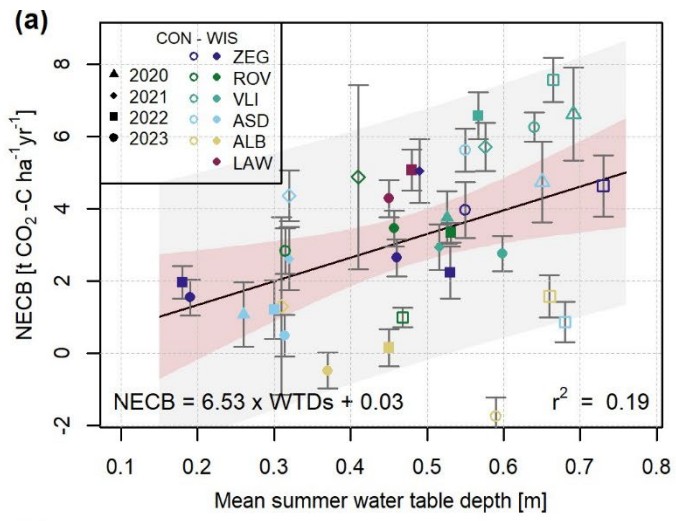

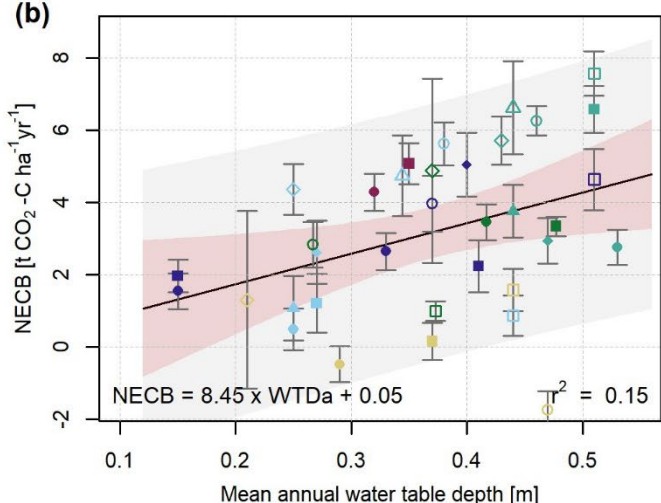

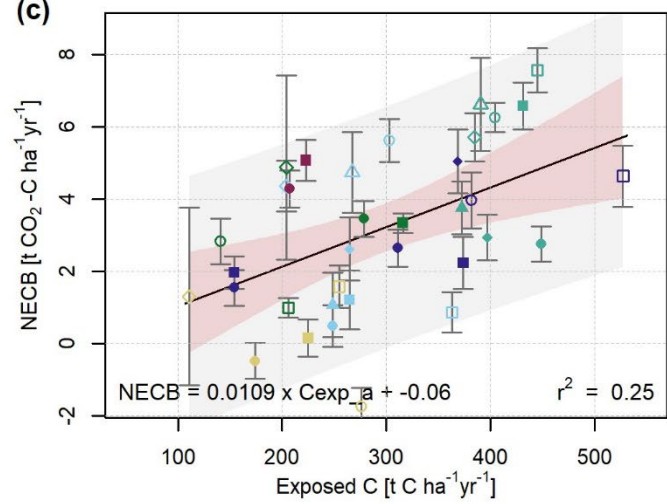

**Figure 7. Mean summer (a) and annual (b) water table depth, and (c) annual exposed carbon with estimated net ecosystem carbon balances (NECB) presented in Table S1. NECB standard deviations are included as error bars. Linear models (Table 2) were fitted on the data and plotted with the 90 % intercept prediction intervals (grey) and 95 % confidence intervals from the linear model estimation (red).**

While simple linear regression is widely applied to fit empirical relations to explain measured NECBs (e.g. Couwenberg et al., 2011; Evans et al., 2021), we tested several alternative linear models for the relation between $WTD_a$ and NECB (Table S2) and inspected the variation in slope and intercept. We included robust linear regression where the weights of outliers are decreased, Deming regression that accounts for observation error estimates, and a linear mixed effect model (LMM) that explicitly models the non-independence in the data (Harrison et al., 2018). The best estimate of the slope of the different linear models ranged from 4.81 (LMM) to 14.35 (Deming model), with the best estimate of the intercept varying from -2.31 (Deming model) to 1.37 (LMM) (Table S2). The best estimate of the slopes of each of these alternative linear regressions was well within the 95 % confidence intervals of the slope estimate of the simple linear regression. The same was true for the intercept that was statistically indistinguishable from zero for all models (Table S2).

### 3.3 Effectiveness of WIS

The average NECB over all the years for the individual plots as function of their average $WTD_a$ and $Cexp_a$ is shown in Fig. 8 a and b, respectively. There is a trend of lower NECBs in the WIS plots compared to the control plots. One notable exception on this trend is ROV. Here, the NECB of the WIS plot exceeds that of the CON plot for the two available years. This location is situated in an area with upward seepage of groundwater, which results in the unintended situation where the WIS mostly drains, rather than infiltrates water. This, in turn, causes a deeper WTD (and higher Cexp) for the WIS plot compared to the CON plot. In this case, a higher NECB in the WIS plot is in line with the expectation based on the relations presented in Table 2.

The question whether WIS is effective in reducing $CO_2$ emissions can be addressed in various ways. For example, one may treat WIS and CON as discrete variables. When including the previously mentioned location with upward seepage (ROV), excluding the location that did not have both WIS and CON sites (LAW), and excluding location-years when either the WIS or CON site did not have data available (i.e. ALB, ROV and ZEG in 2021), the average NECB on WIS sites was 2.26 (16 site-years) t $CO_2$-C ha$^{-1}$ yr$^{-1}$ and on CON sites 3.86 (14 site-years) t $CO_2$-C ha$^{-1}$ yr$^{-1}$. Excluding ROV, the average WIS and CON NECBs were 2.10 (14 site-years) and 4.19 (12 site-years) t $CO_2$-C ha$^{-1}$ yr$^{-1}$, respectively. We find very strong evidence of a reducing effect of WIS on the NECB (LMM: $F_{1,13}=20.82$, $P=5.26 \cdot 10^{-4}$) when excluding ROV, and strong evidence of an effect of WIS ($F_{1,15}=9.50$, $P=0.0075$) when including ROV.

As the primary reason for implementation of WIS is to achieve a shallower WTDs, we can also treat WIS as a continuous variable by considering the effect of WIS on the WTD or Cexp. To do so, we consider the difference in WTD or Cexp

between the WIS and CON plot as explanatory variable, and the difference in NECB between the two plots as effect. This way, also situations where WIS results in a deeper WTD (i.e., contrary to the intended water infiltration effect, as observed in ROV) can be assessed, as based on the relations in Table 2 we expect a higher NECB when WIS deepens the WTD. This comparison is visualized in Fig. 8c and d for $WTD_a$ and $Cexp_a$, respectively. The linear relation displayed in these graphs is the linear relation given in Table 2 and used in Fig. 8a and 8b and seems to fit adequately to the points in the graph. A simple linear regression through these datapoints does not yield a significantly different slope.

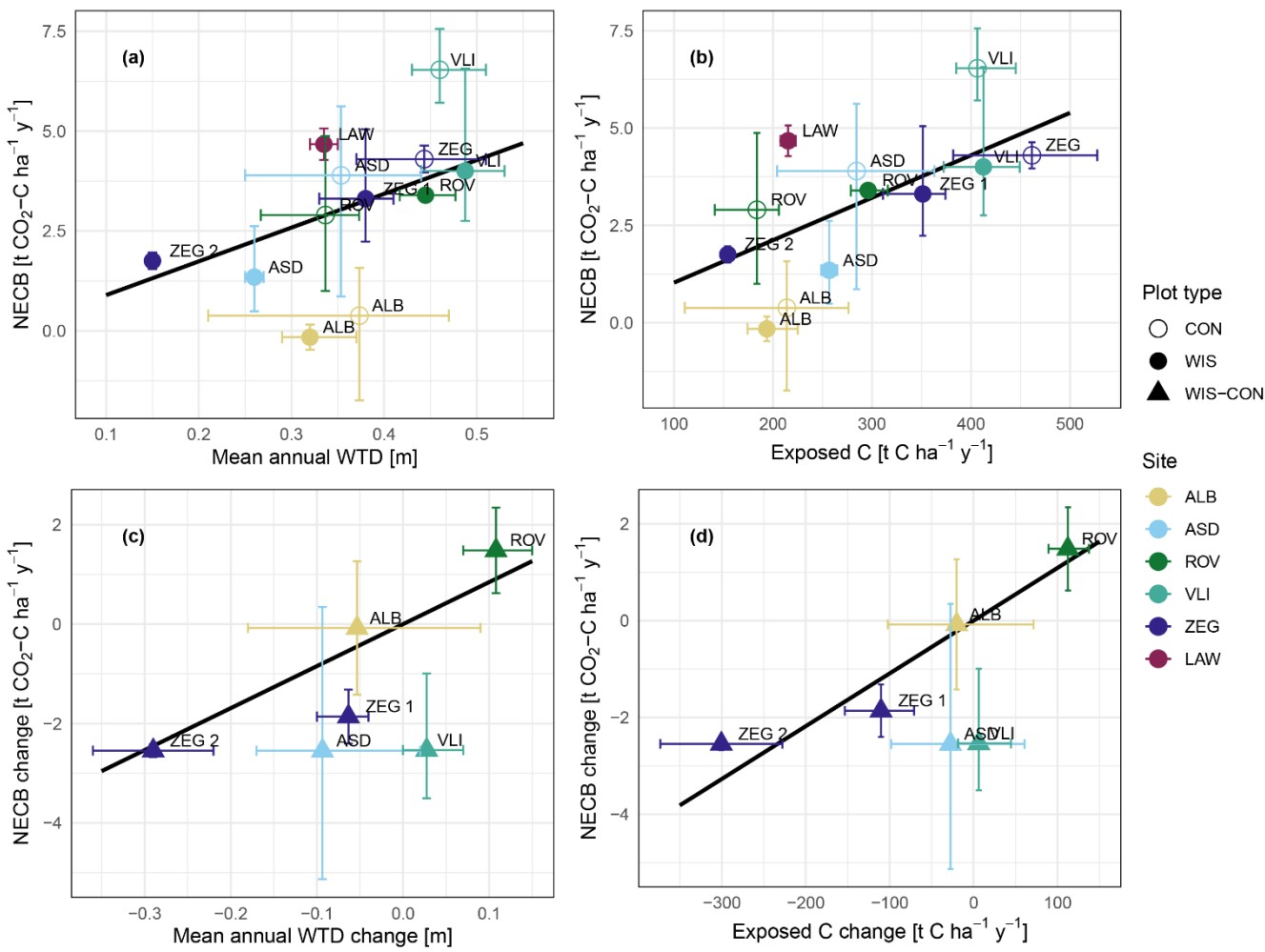

**Figure 8. Averaged annual net ecosystem carbon balance (NECB) per plot as function of (a) averaged annual water table depth (WTD) or (b) averaged annual soil C exposure; (c) averaged annual difference in NECB between WIS and CON (as WIS – CON) sites per location, as function of averaged annual difference in WTD or (d) averaged annual difference in soil C exposure. Black solid lines are the linear model fits of Figure 7. Whiskers indicate minimum and maximum annual values per location or plot.**

To further strengthen this argument, we found no evidence for an effect of WIS on the relationship between NECB and WTD (LMM: $F_{1,24}$=0.79 and $P$=0.38 for $WTD_a$ and $F_{1,23}$=0.52 and $P$=0.48 for $WTD_s$)—as was suggested by Boonman et al. (2022)—nor between NECB and Cexp (LMM: $F_{1,20}$=0.12 and $P$=0.74 for $Cexp_a$, and $F_{1,19}$=0.11 and $P$=0.74 for $Cexp_s$). This implies that the potential impact that WIS may have on environmental factors as soil temperature, nutrient status, electron acceptor availability, or oxygen and dissolved organic C availability fall within the uncertainty of year-to-year and site-to-site variability when using only WTD or exposed C as explanatory variables. This suggests that the linear model fits presented in Table 2 (within the available data ranges) can estimate the reduction in NECB due to a change in WTD or exposed C owing to the implementation of WIS. Therefore, we conclude that if WIS is able to raise the groundwater table substantially, it has a reducing effect on the NECB, based on the paired site comparisons and statistics of fitted WTD-NECB models with slopes exceeding zero in all cases (Table S2).

Previous research showed that WIS resulted in neglectable effects on the NECB (Weideveld et al., 2021), a higher NECB (Tiemeyer et al., 2024) or a mild (Offermanns et al., 2023) to strong (Boonman et al., 2022; Boonman et al., 2024a; van den Akker et al., 2008) reduction in NECB. Here we show that NECB changes in WIS sites are dependent on the actual changes in WTD or exposed C, and that, in some cases, a neglectable or even slightly adverse effect of WIS (as in ALB and ROV, respectively) can be expected if changes in WTD or exposed C are minimal or opposite to the aim of WIS.

Apart from WIS, which typically leads to a moderate WTD increase, more drastic WTD regulation could be implemented to allow paludiculture (Geurts et al., 2019; Martens et al., 2023) or restoration to a full peat growing ecosystem (Nugent et al., 2019) as more effective measures to limit (or even reverse) peat loss (Girkin et al., 2023). When applying these alternative measures, the relation between WTD and NECB that we defined might not be directly applicable due to vegetation differences and a WTD range. Also, at a shallower WTD, other GHG such as $CH_4$ and $N_2O$ might offset reductions in $CO_2$ emissions (Evans et al., 2021; Tiemeyer et al., 2020). Furthermore, a broader perspective on measures (other than GHG emissions) will be necessary since WIS can only reduce peat oxidation to a certain extent, while overall net zero emission is aimed for in 2050. Although we recognize that WIS is an attractive measure to reduce $CO_2$ emissions without changing land-use, we emphasize the need for inclusion of other aspects with respect to the future of managed peatlands. Measures to counteract peat oxidation should always be evaluated from different disciplines and stakeholder perspectives.

### 3.4 NECB estimations and water table depth relationships in perspective

The NECB and WTD observations presented in this study are similar to those of other empirical relations of Evans et al. (2021), Boonman et al. (2022) and Fritz et al. (2017) (Fig. 9, Table S3). However, several other studies found considerably higher emissions from drained peatlands for $WTD_a$ deeper than 0.2 m below surface (Couwenberg et al., 2011; Koch et al., 2023; Tiemeyer et al., 2020). The IPCC emission factors for $CO_2$ emissions from drained organic soils are also higher: the IPCC Wetland Supplement (IPCC, 2014) contains separate emissions factors (EFs) for grassland on nutrient-rich, shallow-

drained (EF1) and nutrient-rich, deep-drained (EF2) organic soils in the temperate climate zone. EF1 applies to a $WTD_a$ of less than 30 cm, whereas EF2 applies to $WTD_a$ of 30 cm and deeper. The NECBs presented in this study for both $WTD_a$ shallower than 30 cm (1.7 t $CO_2$-C ha$^{-1}$ yr$^{-1}$) and deeper than 30 cm (3.8 t $CO_2$-C ha$^{-1}$ yr$^{-1}$) are low compared to EF1 and EF2 (3.6, [95 % CI: 1.8, 5.4] and 6.1 [95 % CI: 5.0 7.3] t $CO_2$-C ha$^{-1}$ yr$^{-1}$, respectively), and fall outside their 95 % confidence intervals. In contrast, our NECBs compare well to those reported by Evans et al. (2021), who used a selection of NECBs obtained across the temperate and boreal regions, including nutrient-poor as well as nutrient-rich sites. Also, multi-year $CO_2$ flux measurements using eddy covariance in the west of the Netherlands lasting from 2005–2008, on sites similar to ours, showed NECB estimations that fall within the prediction intervals of our study, considering an average annual WTD of 0.4–0.5 m (4.2 t $CO_2$-C ha$^{-1}$ yr$^{-1}$, Veenendaal et al. 2007).

Our NECBs also compare well with back-of-the-envelope emissions estimated from land subsidence rates, which range between 2 to 15 mm yr$^{-1}$ in the coastal peat soils in the Netherlands (Hoogland et al., 2012; van den Akker et al., 2008). With an average soil carbon content of $72 \pm 10$ kg m$^{-3}$ at 80 cm depth across our locations (Fig. 2), and assuming that the carbon density profile from the surface up to 80 cm depth is roughly in equilibrium as decomposition due to drainage for agricultural use has been ongoing for at least 50 years (but at most sites over multiple centuries; e.g. Erkens et al., 2016), we infer emissions ranging between 1.5 and 11 t $CO_2$-C ha$^{-1}$ yr$^{-1}$. These subsidence-derived estimates correspond well to our estimated NECBs (Fig. 9) and thus strengthen the presented approach to derive annual NECBs.

It is notable that our NECB estimates as well as the slope of the $WTD_a$-NECB relationship are on the lower side of those reported by Tiemeyer et al. (2020) and Koch et al. (2023) (Fig. 9). There are several potential explanations for differences between our results and those of others. First, magnitudes of estimated NECBs and different NECB-WTD relationships may be related to differences in landscape, peat soil characteristics, peat decomposition-state and land use history and practises (e.g. Evans et al. 2021, Tiemeyer et al. 2016). For example, in contrast to the aforementioned studies, the measurements presented in this paper are exclusively conducted on coastal peatlands, which often have a relatively high WTD, limited drainage, clay cover, and the ability for meticulous water management, whereas the studies mentioned earlier are compiled from measurements in a larger variety of peatlands with few data from coastal peatlands. The $WTD_a$-NECB relationships of Tiemeyer et al. (2020), Koch et al. (2023), and Evans et al. (2021) are based on NECBs from sites that widely differ in land use and are based on a $WTD_a$ range that differs from the one where our relationship was fitted on, as particularly Tiemeyer et al. (2020), but also Koch et al. (2023) contain NECBs from sites where the $WTD_a$ lies far outside our measured range. These factors could affect the nature of the relationship as at least some aspects of land use may have effects independent of those of $WTD_a$ (Evans et al. 2021) and since deepening of the $WTD_a$ likely has a finite effect on the oxygen penetration depth in the peat soil (Boonman et al., 2024b). Another factor that can affect the magnitude of the NECB and its relationship with WTD is the clay cover that is typical of the Dutch coastal peatlands (Koster et al. 2018). A clay cover limits the thickness of the layer of peat that is exposed to oxygen, and hence, limits mineralisation (Jansen et al., 2009). Additionally, a clay cover

and mixtures of clay with peat may suppress mineralisation and related $CO_2$ emissions from peat (e.g. Deru et al., 2018) via 1) via clay-labile carbon complexation that restricts degradation of the organic matter by microorganisms (Hassink et al., 1997; Torres-Sallan et al., 2017; Rumpel et al., 2015); 2) restricting oxygen transport to organic matter by decreasing soil pore sizes and increasing soil water content (Balogh et al., 2011); and 3) by altering interactions with microorganisms and their enzymes (Turner et al., 2014). The presence of clay cover may thus contribute to observed differences between the magnitude of our NECBs and its relationship with WTD as compared to those from other European countries. Last, differences could be related to methodological issues, such as potential biases due to a changing microclimate in automated chambers (Maier et al., 2022; Oestmann et al., 2022; Yao et al., 2009), gap-filling uncertainties/choices (Liu et al., 2022) and choices in data-handling (Hoffmann et al., 2015; Shi et al., 2022). Different methods to determine NECBs have their own pros and cons (Liu et al., 2022) and should be used complementary as much as possible.

Although a sigmoidal function was used to model the $WTD_a$-NECB relationship on the entire dataset of Tiemeyer et al. (2020) and Koch et al. (2023), within our measured $WTD_a$ range, a (pseudo)linear trend is evident in the subset of their data. To enable a fairer comparison between the $WTD_a$-NECB relationships based on our data and those from literature, we selected a subset of data from the Evans et al. (2021), Tiemeyer et al. (2020) and Koch et al. (2023) syntheses where $WTD_a$ was within the range of our measurements (i.e., $WTD_a$ not more than 5 cm outside of our $WTD_a$ range; Fig. 9b). In addition, we only selected data from sites with similar land use as our sites—i.e., only grassland sites from Evans et al. (2021), only permanent or rotational grassland sites from Koch et al. (2023), and only sites where *Lolium perenne* was among the dominant species from Tiemeyer et al. (2020). By including data source as an interaction term with $WTD_a$ in our linear mixed-effects model (LMM), we can isolate the $WTD_a$ effect from potential differences in the slope or intercept of the compared relationships. As such we can analyse whether combining our dataset with one from literature adds evidence for an effect of $WTD_a$ on the NECB (i.e., increase effect variance relative to error variance) and determine the evidence for a difference in slope and intercept between our $WTD_a$-NECB relationship and that of a given dataset from literature. When comparing our $WTD_a$-NECB relationship with the one based on the grassland sites (11 data points) of Evans et al. (2021), we found moderate evidence for an effect of $WTD_a$ on the NECB (LMM: $F_{1,32}=6.88$; $P=0.019$) and no evidence for an effect of the data source (i.e., Evans et al. 2021 vs this study) on the slope (LMM: $F_{1,32}=1.02$; $P=0.32$) and intercept (LMM: $F_{1,29}=1.70$; $P=0.20$) of the $WTD_a$-NECB relationship. On the contrary, when we did this analysis for subset of data from Tiemeyer et al. (2020) (12 data points), we found no evidence for an effect of $WTD_a$ on the NECB (LMM: $F_{1,40}=1.09$; $P=0.30$) and no evidence for an effect of the data source on the slope (LMM: $F_{1,40}=0.06$; $P=0.81$) and the intercept (LMM: $F_{1,30}=1.21$; $P=0.28$) of the $WTD_a$-NECB relationship. When comparing the $WTD_a$-NECB relationship based on the subset of Koch et al. (2023) with our relationship, there also was no evidence for an effect of $WTD_a$ on the NECB (LMM: $F_{1,22}=0.40$; $P=0.54$), no evidence for an effect of the data source on the slope (LMM: $F_{1,22}=2.35$; $P=0.14$) and strong evidence for an effect of data source on the intercept LMM: $F_{1,19}=9.27$; $P=0.0067$) of the $WTD_a$-NECB relationship. Combining our data with the subset of Evans et al. (2021) in the LMM resulted in stronger evidence for an effect of $WTD_a$ ($P=0.019$) than when

testing the effect of WTD$_a$ for each data set independently (LMM: F$_{1,32}$=1.96; *P*=0.17 and LM: F$_{1,9}$=5.78; *P*=0.040 for our study and the subset of Evans et al. (2021), respectively). On the contrary, combining our data with the subset of Tiemeyer et al. (2020) or Koch et al. (2023) weakened the evidence for an effect of WTD$_a$ as compared to only using our data in the LMM. These findings suggest that the relationship between WTD$_a$ and NECB may not be consistent across all drained

peatlands in use as grassland or under all environmental conditions. For example, it may imply that—compared to our dataset and the one of Evans et al. (2021)—the German and Danish sites show greater variation in NECBs independent of WTD$_a$, that may result from greater variation in land-use intensity or properties of the peat soil (see previous paragraph). Lastly, our results may imply that the various WTD-NECB relationships as well as magnitudes of NECBs are sensitive to

methodological differences, as the data of Tiemeyer et al. (2020) and Koch et al. (2023) are based on campaign-wise measurements during daytime with manual chambers, while our data (automated chambers) and data of Evans et al. (2021) (eddy covariance) were collected with much higher temporal cover, which reduces the extent (and thereby uncertainty) of gap-filling and the need to predict nighttime $CO_2$ fluxes based on daytime measurements with opaque chambers. Future research should focus on comparing and validating the various methodologies—including effects of the extent of gap-

filling—as well as causes of potential regional physical variation in NECB magnitudes.

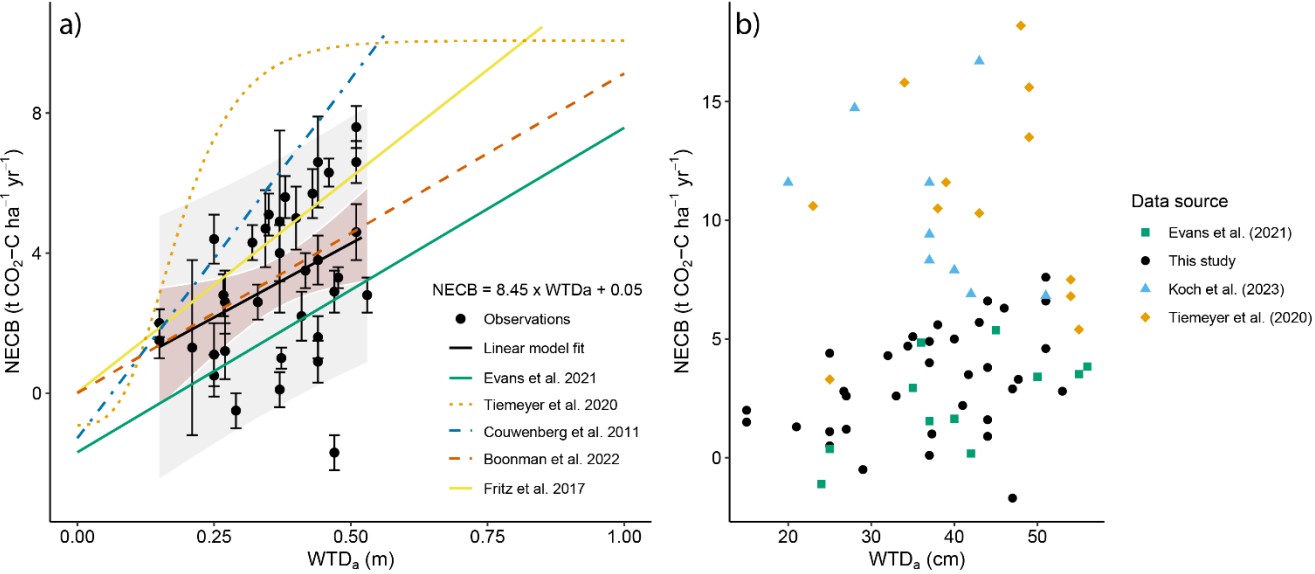

**Figure 9. (a) Fitted linear model of measured mean annual water table depth (WTD$_a$) and net ecosystem carbon balance (NECB) of Table 2 compared to other empirical relations. NECB standard deviations are included as error bars, and the linear model is plotted with a 90 % prediction interval (grey shading) and a 95 % confidence interval for the linear fit (red shading). An overview**

**of plotted models is presented in Table S3. Please note that Koch et al. (2023) found an identical fit as Tiemeyer et al. (2020), which therefore is not separately displayed. (b) WTD$_a$ and NECB estimates of this study and from literature. Only sites with similar land use (grassland) and a WTD$_a$ within the range of our measurements (i.e. WTD$_a$ not more than 5 cm outside of our WTD$_a$ range) were selected for fair comparison.**

## 3.5 Landscape-scale emissions

*Upscaling emissions*

To upscale emission estimates to those at the regional and national level, it is important that our results are included in mechanistic models that contain (geographic) data to account for things like spatial heterogeneity of peat types, peat depth, hydrology, year-to-year variation in weather conditions, type of measure (e.g. passive or active WIS) and management. Efforts are already being made to enable such upscaling of results using a multi-model ensemble (Erkens et al., 2022). Nevertheless, it becomes clear from our results that the application of WIS alone will be insufficient to achieve the targeted 95% reduction of emissions in 2050. Hence, to achieve the emission reduction target, WIS can fit in as a (temporary) measure combined with more drastic rewetting measures. Lastly, when upscaling emissions and effects of management at the landscape scale, one should not only consider the direct land-atmosphere fluxes, but also those from other landscape elements, such as ditches, that are affected by the mineralisation and management of the peat soil (as discussed below).

*Waterborne export*

Our NECBs determined via chamber measurements do not account for carbon fluxes via runoff, lateral subsurface flow and emission of $CH_4$, CO and volatile organic carbon. While emission of the latter three gases is likely negligible (e.g. Weideveld et al., 2021; Faubert et al., 2011; and Aben et al. [unpublished CO data]), carbon losses via runoff, erosion and lateral subsurface flow can be significant (Evans et al., 2016). This carbon is partly mineralized and emitted to the atmosphere in surrounding ditches as well as further downstream in the hydrological system.

*Ditch emissions*

Carbon and GHG emissions from ditches in managed peatlands can be substantial and are important on the landscape scale (Vermaat et al., 2011; Schrier-Uijl et al., 2014; Peacock et al., 2017; Piatka et al., 2024). For example, GHG emissions ($CH_4$, $CO_2$, $N_2O$) from ditches in drained peatlands in the north of the Netherlands were estimated to be 4.8 times larger on a per area basis than those of the terrestrial peat, forming an estimated 20% of landscape-scale emissions (Hendriks et al. 2024). Thus, to quantify carbon and GHG emissions from drained peatlands on the landscape scale, it is crucial to include emission estimates from ditches and downstream waters. Care should be taken not to include these emissions twice, as waterborne carbon export from the soil forms part of the carbon emission from ditches where the waterborne carbon export ends up.

*Management*

Similarly, effects of measures on waterborne exports and ditch emissions need to be quantified, as subsurface and shallow surface drains in managed peatlands likely stimulate losses of dissolved and particulate carbon as well as dissolved GHGs, sulphate and nutrients (Uusitalo et al., 2001; Vermaat et al., 2016; Kladivko et al., 2021; Pickard et al., 2022). The latter two can stimulate anaerobic mineralisation of the organic ditch sediment while simultaneously contributing to external and internal eutrophication (Smolders et al., 2006) that in turn stimulates GHG emission (Beaulieu et al., 2019).

## 4. Conclusions

We presented the results of a novel and unprecedented $CO_2$ emission monitoring network for peatlands under intensive agricultural use (grassland) in the Netherlands, using automated transparent chambers. High-frequent measurements of $CO_2$ fluxes and supporting data (e.g. water table depth [WTD] and weather) provided us with up to four years of near continuous, high frequency measurements for twelve sites in the Netherlands, which we used to determine the annual net ecosystem carbon budget (NECB). The sites consisted of plots where water infiltration systems (WIS) were implemented, combined with nearby control plots. For the ranges in WTD considered in this study, we found a linear relation between NECB and annual (as well as summer) WTD as was presented in literature before. However, a stronger relation was found between NECB and carbon exposure (Cexp), expressed as the amount of available soil carbon above the WTD. We therefore propose to use the carbon exposure rather than the WTD as a proxy for the NECB. Still, substantial variation in NECB could not be explained by these variables, which deserves attention in future analyses. The WIS studied were proven to be effective in decreasing peatland $CO_2$ emissions in case they function as intended (i.e. raising the WTD). We found no evidence for an effect of WIS on the slope of the relation between NECB and WTD, nor on the slope of the relation between NECB and Cexp. The magnitude of our NECBs and the slope of the WTD-NECB relationship agreed well with some studies, but not with all. This is potentially explained by regional differences in physical geographic setting, peat type and land-use history and water management, and/or by methodological differences and warrants further analysis. The large site-to-site and year-to-year variation calls for continuation of near-continuous, high-frequency measurements to further improve our understanding of the drivers of greenhouse gas emissions from peatlands in agricultural use.

### Data availability

Data of annual carbon budgets, NEE, GPP, $R_{eco}$, harvest, water table depth and exposed carbon can be found in Table S1. Other data, such as timeseries of $CO_2$ fluxes, are not yet publicly available due to ongoing research but are available from the corresponding author on reasonable request.

### Author contributions

MvdB and GE led the design of the study with contributions from RA, DvdC, JB and YvdV. RA, DvdC, JB, SP and CB processed data from chamber measurements and ancillary measurements. BV contributed soil C profile data. MvdB, SP and DvdC performed gapfilling of the data. RA, DvdC and JB performed the statistical analyses. RA, DvdC and JB led the manuscript writing. GE, YvdV, CB, MvdB and BV contributed to revisions of the manuscript.

**Competing interests**

The authors declare that they have no conflict of interest.

**Acknowledgements**

This research was funded by the Netherlands Research Programme on Greenhouse Gas Dynamics in Peatlands and Organic Soils (NOBV) and (co-)funded by the WUR internal program KB34 Towards a Circular and Climate Neutral Society (2019-2024), project KB-34-002-005 (Reversing declining soils mitigating climate innovation in peatland management). We thank our (former) colleagues within the NOBV project for maintaining the field sites, providing technical support and managing remote data accessibility. We also thank the landowners for allowing us to conduct this research on their fields.

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
