# Peer review of "CO2 emissions of drained coastal peatlands in the Netherlands and potential emission reduction by water infiltration systems"

_EGUsphere, 2024_

## Author Comment (AC2)

Anonymous Referee #1, 19 Mar 2024

*We thank the reviewer for the positive and constructive evaluation of our manuscript. All comments are 1-by-1 addressed below and will help us greatly to further improve our study. Note that we found a slight inconsistency in the calculation of several of the NECBs and average groundwater tables. This means that a revised version of the manuscript will contain some updated values of NECBs and groundwater tables, and relationships with NECBs that have been adjusted accordingly. However, these changes do not affect the overall conclusions or the interpretation of our results.*

The authors present a comprehensive data set on $CO_2$ fluxes and carbon balances for a series of managed Dutch peatlands with the aim to address possible mitigation effects from subsurface water infiltration systems. This is an excellent and well elaborated study in terms of applied techniques in the field and in the statistical section, as well as in the depth of data analysis and interpretation. I strongly recommend the preprint to be published as full paper in Biogeosciences after consideration of the comments below.

*Thank you for the kind words and positive assessment of our manuscript.*

General comments

Abstract. Please provide an estimate on the potential of WIS to reduce peatland emissions in the Netherlands.

*We will include this estimate in the abstract of our revised manuscript.*

Line 51. Please explore in the discussion whether the target of -95% by 2050 could be reached with WIS.

*In response to this comment and several other comments we will add a paragraph to the discussion on how we envision our results can be used to derive regional or national emission estimates. In this paragraph we discuss carbon fluxes that we did not measure, the importance of ditches as source of emissions, the use of mechanistic models to upscale the presented NECBs to regional fluxes taking into account the heterogeneity of e.g. peat types, hydrology and management. In this section we also add a discussion on the potential of WIS to reach the targeted emission reduction. From our results it is clear that WIS is likely to reduce emissions when WIS lead to substantial higher groundwater tables in summer, but that it will not yield near carbon-neutral conditions.*

*Suggested new paragraph:*

**3.5 Landscape-scale emissions**

*Upscaling emissions*

*To upscale emission estimates to those at the regional and national level, it is important that our results are included in mechanistic models that contain (geographic) data to account for things like spatial heterogeneity of peat types, peat depth, hydrology, year-to-year variation in weather conditions, type of measure (e.g. passive or active WIS) and management. Efforts are already being made to enable such upscaling of results using a multi-model ensemble (Erkens et al., 2022). Nevertheless, it becomes clear from our results that the application of WIS alone will be insufficient to achieve the targeted 95% reduction of emissions in 2050. Hence, to achieve the emission reduction target, WIS can fit in as a (temporary) measure combined with more drastic rewetting measures. Lastly, when upscaling emissions and effects of management at the landscape scale, one should not only consider the direct land-atmosphere fluxes, but also those from other landscape elements, such as ditches, that are affected by the mineralisation and management of the peat soil (as discussed below).*

*Waterborne export,*

Our NECBs determined via chamber measurements do not account for carbon fluxes via runoff, lateral subsurface flow and emission of $CH_4$, CO and volatile organic carbon. While emission of the latter three gases is likely negligible (e.g. Weideveld et al., 2021; Faubert et al., 2011; and Aben et al. [unpublished CO data]), carbon losses via runoff, erosion and lateral subsurface flow can be significant (Evans et al., 2016). This carbon is partly mineralized and emitted to the atmosphere in surrounding ditches as well as further downstream in the hydrological system.

*Ditch emissions*

Carbon and GHG emissions from ditches in managed peatlands can be substantial and are important on the landscape scale (Vermaat et al., 2011; Schrier-Uijl et al., 2014; Peacock et al., 2017; Piatka et al., 2024). For example, GHG emissions ($CH_4$, $CO_2$, $N_2O$) from ditches in drained peatlands in the north of the Netherlands were estimated to be 4.8 times larger on a per area basis than those of the terrestrial peat, forming an estimated 20% of landscape-scale emissions (Hendriks et al. 2023). Thus, to quantify carbon and GHG emissions from drained peatlands on the landscape scale, it is crucial to include emission estimates from ditches and downstream waters. Care should be taken not to include these emissions twice, as waterborne carbon export from the soil forms part of the carbon emission from ditches where the waterborne carbon export ends up.

*Management*

Similarly, effects of measures on waterborne exports and ditch emissions need to be quantified, as subsurface and shallow surface drains in managed peatlands likely stimulate losses of dissolved and particulate carbon as well as dissolved GHGs, sulphate and nutrients (Uusitalo et al., 2001; Vermaat et al., 2016; Kladivko et al., 2021; Pickard et al., 2022). The latter two can stimulate anaerobic mineralisation of the organic ditch sediment while simultaneously contributing to external and internal eutrophication (Smolders et al., 2006) that in turn stimulates GHG emission (Beaulieu et al., 2019).

Line 102. I suggest to explore the presence of clay on top of the peat layer in the introduction – how frequent are these situations in the Netherlands, how may a clay horizon affect the $CO_2$ balance when the water table is raised etc. Many readers may not be familiar with this type of soil.

*Thank you for this suggestion. We will add the following information on the relative occurrence of clay-covered peat soils to the introduction:*

"Due to deltaic and coastal conditions 17 % and 36 % of coastal peatlands in the Netherlands are covered by a thick (40–80 cm) and thin (<40 cm) clay cover, respectively (Jansen et al., 2009)."

*Furthermore, we will add a section on the potential effects of clay cover on (drainage-related) $CO_2$ emissions to the existing section of results & discussion that covers potential explanations for differences between our results and those from other studies:*

"Another factor that can affect the magnitude of the NECB and its relationship with WTD is the clay cover that is typical of the Dutch coastal peatlands (Koster et al. 2018). A clay cover limits the thickness of the layer of peat that is exposed to oxygen, and hence, limits mineralisation (Jansen et al., 2009). Additionally, a clay cover and mixtures of clay with peat may suppress mineralisation and related $CO_2$ emissions from peat (e.g. Deru et al., 2018) via 1) via clay-labile carbon complexation that restricts degradation of the organic matter by microorganisms (Hassink et al., 1997; Torres-Sallan et al., 2017; Rumpel et al., 2015); 2) restricting oxygen transport to organic matter by decreasing soil pore sizes and increasing soil water content (Balogh et al., 2011); and 3) by altering interactions with microorganisms and their enzymes (Turner et al., 2014). The presence of clay cover may thus contribute to observed differences between the magnitude of our NECBs and its relationship with WTD as compared to those from other European countries."

Methods: Do sites contain inorganic carbon? If so, this should be mentioned at some point.

*Inorganic carbon is not added to the experimental fields and is also not widely present in the soil profile. Inorganic carbon is present in the soil water phase as dissolved $CO_2$ and $HCO_3^-$, mostly originating from peat decomposition. For the NECB calculation, we assume that changes in inorganic carbon in the soil profile including the water phase on a yearly basis (1-Jan–1-Jan) are small compared to the GPP and Reco fluxes and mostly fall within the uncertainty bounds of the overall fluxes. We will add this information to the methods section.*

$CO_2$ measurements. At site Zegveld the control is compared to two different chamber types; how does this influence the interpretation of results?

*We made a large effort to harmonise the procedures followed with both chamber types. Moreover, we did not find strong effects of chamber type on the yearly carbon budgets. This is best seen in Fig. 8b where both chamber types (ZEG2, ASD, VLI, and LAW versus ALB, ZEG1 and ROV) give average NECBs above and below the average relationship between exposed carbon and water level. ZEG is the only site where "control" and "WIS treatment" have been measured with 2 different chamber types. In figure 8d we do see that ZEG2 has the largest change in exposed carbon of all sites. Hower the regression line that is shown is the regression line of figure 7c (i.e. not determined on the points of figure 8d). The observation that both ZEG1 (passive water infiltration system) and ZEG2 (active water infiltration system) follow the overall relationship derived from all yearly values, gives us extra confidence that differences between chamber types are small. We additionally ran a linear mixed-effects model testing for an effect of chamber type on the relation between annual exposed carbon and NECB, which resulted in no evidence for such an effect ($F_{1,12}$=0.03; $P$=0.87).*

Lines 220ff. Please provide a (supplementary) figure or an error estimate for predicting Reco from temperature for the individual sites.

*We will provide a figure with the error estimate for the predicted daytime Reco based temperature and the nighttime data, which will be based on the error estimate of E0.*

Line 228. Sentence starting with 'We partitioned' belongs to the beginning of para 2.2.4.

*Thank you. We will correct this in the revised manuscript.*

Line 310. Useful approach.

*Thank you.*

Line 413. This is a very relevant finding and should be presented also in the abstract.

*We will add this to the abstract.*

Line 419 ff. Authors present results for alternative regression models but it is not explained what to do with this finding.

*The alternative regression models are mainly meant as an eye-opener to the community to generate awareness that the type of regression that is chosen can substantially affect the best estimate and confidence intervals of slopes and intercepts. Since there can be good arguments to choose any of the presented alternative regression models, we refrain from making any recommendations here. The simple linear regression receives most attention in our manuscript as it is compared to other relationships in literature that are based on simple linear regression.*

General: The MS would benefit from showing the year-round GW measurements in a supplementary figure (and not only aggregated as in Table S1) as this would visualize the effect of WIS.

*We will add Figure S2 that contains timeseries graphs showing hourly values of the average water table depth (WTD) for the control (CON) and water infiltration system (WIS) plot for each of the studied locations.*

Points to be included in the discussion section:

Line 297. DOC and other pathways were not considered, which is acceptable given the efforts to estimate those terms. However, leaving them out from the overall budget calculation implies that the observed net C losses from the studied systems might be actually higher. This should be discussed later, together with literature estimates from similar systems (if available), denoting the possible magnitude of this effect.

*This is indeed an important topic, not only from the viewpoint of the magnitude of C emission related to this waterborne C export, but also considering a potential difference in waterborne C export between control and treatment sites that needs to be addressed. The suggested new paragraph 3.5 (see our reply on page 1) discusses this matter in detail. Note that we did not include estimates from similar systems, since we are not aware of any studies of Dutch coastal peatlands that quantified waterborne C losses.*

Data in Fig. 6 show that manure affects the NECB. How much manure is produced from the harvests per site and how much would therefore be available to improve the NECB? This is more than a theoretical question as a the effect of manure on the ecosystem's C budget is now accounted for elsewhere.

*The main objective of this study was to quantify C budgets of the peat soil, rather than C budgets at the ecosystem scale (i.e., including farm-scale emissions). Hence, we did not investigate how much of the harvest at each site was fed to the cows and how much manure was produced from that. Regarding our studied plots, manure was only applied where necessary (i.e., the organic farm of location Aldeboarn [paragraph 2.4]).*

You discuss why are NEE/NECB results are so different to Tiemeyer et al. 2020. Could the clay layer also play a role? The SOC within the clay layer may be less prone to decomposition as unprotected peat, for example.

*The clay layer may indeed contribute to differences between our results and those of Tiemeyer et al. 2020. We will add a discussion on this matter to the main text (see our reply on page 2 for the contents of this discussion).*

Technical remarks

Line 68. Please update references Buzacott and van den Berg.

*We will update the references in question.*

Line 68. Sentence not complete. They mean a change from conventional agricultural land use towards what?

*We meant moving away from conventional agriculture to alternatives, such as wet agriculture (paludiculture) or rewetting for nature restoration. We will clarify this sentence in the revised manuscript.*

Line 75. Please add: 'as the hydraulic conductivity of degraded peat soils is…'

*We will add 'degraded' to the sentence.*

Figure 2. I suggest to replace the grey bars for the WTD by blue bars or another colour providing better visibility. The figure provides C concentrations in kgC/m3; this is a C density, a C concentration would be kgC/kg soil.

*We will revise this figure and the description accordingly.*

Figure 4. Please revise so that either dark and light red is replaced by a different colour.

*We will change the figure so that it has a two color gradient, instead of dark red to white.*

Line 328. Please move first parenthesis before '2022'.

*We will remove the parentheses.*

Figure 9b. I suggest to use also different symbol types, not only different colours in this plot.

*We will use different symbols in Figure 9b. We will also use additional line types in Figure 9a and a color blind-proof color palette to make lines and symbols distinguishable for the various kinds of color blindness.*

---

## Author Comment (AC3)

Anonymous Referee #2, 20 Apr 2024

*We thank the reviewer for the positive and constructive evaluation of our manuscript. All comments are 1-by-1 addressed below and will help us greatly to further improve our study. Note that we found a slight inconsistency in the calculation of several of the NECBs and average groundwater tables. This means that a revised version of the manuscript will contain some updated values of NECBs and groundwater tables, and relationships with NECBs that have been adjusted accordingly. However, these changes do not affect the overall conclusions or the interpretation of our results.*

The manuscript is based on an impressive dataset and is clearly written and good English language. This kind of information on the effectiveness of GHG mitigation measures on peat soils and methods to estimate the emissions based on environmental variable is urgently needed. Comparison of the WTD-NECB relationship in different datasets (Fig. 9) was especially interesting in this manuscript. I have only minor comments.

*Thank you for the kind words and positive assessment of our work.*

Title: I'm not sure if the title should start with "Using automated transparent chambers to quantify…" as this was not a methodology-oriented paper to my opinion. I would stress the large dataset by including the number of sites and probably "continuous measurements" in the title if needed. Could it be: $CO_2$ emission reduction potential of water infiltration systems at six drained coastal peatland sites in the Netherlands.

*Thank you for the suggestion. We will remove the automated transparent chamber part of the title and revise it to: "$CO_2$ emissions of drained coastal peatlands in the Netherlands and potential emission reduction by water infiltration systems"*

I think the infiltration system you used is also called submerged drainage. If this is true, please also include this term in the methods section to make it clearer for readers who are less familiar with (Dutch) drainage systems.

*This is true. We will add this clarification in the revised manuscript.*

In order to understand the functioning of the WIS, a figure on the WTD variation within a year would be useful.

*We will add a Figure S2 that contains timeseries graphs showing hourly values of the average water table depth (WTD) for the control (CON) and water infiltration system (WIS) plot for each of the studied locations.*

Table 1: should the title in the 6th column be "ditch WTD", not "aim".

*We will revise this to "targeted ditch WT" to reflect that the values in this column are not measured values but ditch water tables targeted by the water authority.*

Line 347: is-->was

*We will correct this accordingly.*

Line 370: add also the mean value (all sites) for NECB.

*We will add the mean value of the NECB across all sites to the revised manuscript.*

Line 412: This kind of observations on the proportion of C lost annually do not widely exist. You could add it to the abstract.

*We will add this to the abstract.*

Line 478: Paludiculture is not a water management system but a cultivation system. You could even use WIS to raise the WT for paludiculture (if possible to raise the WTD to 20 cm). I suggest revising this sentence e.g. to: Apart from WIS, typically leading to moderate WTD increase, more efficient WTD regulation could be implemented to allow paludiculture (Geurts et al., 2019; Martens et al., 2023) or restoration to a full peat growing ecosystem (Nugent et al., 2019).

*Thank you for the suggestion. We propose the following revised text:*

*"Apart from WIS, which typically leads to a moderate WTD increase, more drastic WTD regulation could be implemented to allow paludiculture (Geurts et al., 2019; Martens et al., 2023) or restoration to a full peat growing ecosystem (Nugent et al., 2019) as more effective measures to limit (or even reverse) peat loss (Girkin et al., 2023)."*

---

## Author Response (AR1)

*Dear Editor and reviewers*

*We thank both reviewers for the positive and constructive evaluation of our manuscript. All comments are 1-by-1 addressed below and helped us greatly to further improve our study. Note that we found a slight inconsistency in the calculation of several of the NECBs and average groundwater tables. This means that the revised version of the manuscript contains some updated values of NECBs and groundwater tables, and relationships with NECBs that have been adjusted accordingly. However, these changes do not affect the overall conclusions or the interpretation of our results. We also changed the colours, symbols and line types of the various graphs to make data better distinguishable for the various types of colour blindness.*

*Note that all line number references below refer to the manuscript containing track-changes, rather than the clean version.*

Anonymous Referee #1, 5 Jun 2024

The authors present a comprehensive data set on $CO_2$ fluxes and carbon balances for a series of managed Dutch peatlands with the aim to address possible mitigation effects from subsurface water infiltration systems. This is an excellent and well elaborated study in terms of applied techniques in the field and in the statistical section, as well as in the depth of data analysis and interpretation. I strongly recommend the preprint to be published as full paper in Biogeosciences after consideration of the comments below.

*Thank you for the kind words and positive assessment of our manuscript.*

General comments

Abstract. Please provide an estimate on the potential of WIS to reduce peatland emissions in the Netherlands.

*We now include an estimate in the abstract (line 29).*

Line 51. Please explore in the discussion whether the target of -95% by 2050 could be reached with WIS.

*In response to this comment and several other comments we added a paragraph to the discussion (paragraph '3.5 Landscape-scale emissions') that discusses how we envision our results can be used to derive regional or national emission estimates. In this paragraph we discuss carbon fluxes that we did not measure, the importance of ditches as source of emissions, the use of mechanistic models to upscale the presented NECBs to regional fluxes taking into account the heterogeneity of e.g. peat types, hydrology and management. In this section we also add a discussion on the potential of WIS to reach the targeted emission reduction. From our results it is clear that WIS is likely to reduce emissions when WIS lead to substantial higher groundwater tables in summer, but that it will not yield near carbon-neutral conditions.*

Line 102. I suggest to explore the presence of clay on top of the peat layer in the introduction – how frequent are these situations in the Netherlands, how may a clay horizon affect the $CO_2$ balance when the water table is raised etc. Many readers may not be familiar with this type of soil.

*Thank you for this suggestion. We added the following information on the relative occurrence of clay-covered peat soils to the introduction (line 50–52):*

"Due to deltaic and coastal conditions 17 % and 36 % of coastal peatlands in the Netherlands are covered by a thick (40–80 cm) and thin (<40 cm) clay cover, respectively (Jansen et al., 2009)."

*Furthermore, we added a section on the potential effects of clay cover on (drainage-related) $CO_2$ emissions to the existing section of results & discussion that covers potential explanations for differences between our results and those from other studies (line 551–560):*

"Another factor that can affect the magnitude of the NECB and its relationship with WTD is the clay cover that is typical of the Dutch coastal peatlands (Koster et al. 2018). A clay cover limits the thickness of the layer of peat that is exposed to oxygen, and hence, limits mineralisation (Jansen et al., 2009). Additionally, a clay cover and mixtures of clay with peat may suppress mineralisation and related $CO_2$ emissions from peat (e.g. Deru et al., 2018) via 1) via clay-labile carbon complexation that restricts degradation of the organic matter by microorganisms (Hassink et al., 1997; Torres-Sallan et al., 2017; Rumpel et al., 2015); 2) restricting oxygen transport to organic matter by decreasing soil pore sizes and increasing soil water content (Balogh et al., 2011); and 3) by altering interactions with microorganisms and their enzymes (Turner et al., 2014). The presence of clay cover may thus contribute to observed differences between the magnitude of our NECBs and its relationship with WTD as compared to those from other European countries."

Methods: Do sites contain inorganic carbon? If so, this should be mentioned at some point.

*Inorganic carbon is not added to the experimental fields and is also not widely present in the soil profile. Inorganic carbon is present in the soil water phase as dissolved $CO_2$ and $HCO_3^-$, mostly originating from peat decomposition. For the NECB calculation, we assume that changes in inorganic carbon in the soil profile including the water phase on a yearly basis (1-Jan–1-Jan) are small compared to the GPP and Reco fluxes and mostly fall within the uncertainty bounds of the overall fluxes. We added this information to the methods section.*

$CO_2$ measurements. At site Zegveld the control is compared to two different chamber types; how does this influence the interpretation of results?

*We made a large effort to harmonise the procedures followed with both chamber types. Moreover, we did not find strong effects of chamber type on the yearly carbon budgets. This is best seen in Fig. 8b where both chamber types (ZEG2, ASD, VLI, and LAW versus ALB, ZEG1 and ROV) give average NECBs above and below the average relationship between exposed carbon and water level. ZEG is the only site where "control" and "WIS treatment" have been measured with 2 different chamber types. In figure 8d we do see that ZEG2 has the largest change in exposed carbon of all sites. Hower the regression line that is shown is the regression line of figure 7c (i.e. not determined on the points of figure 8d). The observation that both ZEG1 (passive water infiltration system) and ZEG2 (active water infiltration system) follow the overall relationship derived from all yearly values, gives us extra confidence that differences between chamber types are small. We additionally ran a linear mixed-effects model testing for an effect of chamber type on the relation between annual exposed carbon and NECB, which resulted in no evidence for such an effect ($F_{1,12}=0.03$; $P=0.87$).*

Lines 220ff. Please provide a (supplementary) figure or an error estimate for predicting Reco from temperature for the individual sites.

*After reconsidering the error estimation for Reco based on the temperature response (eq.3), we conclude that it is not relevant for the paper to provide this extra data. The only reason why Reco is calculated is to gap-fill Reco and GPP separately. The procedure for estimating the gap-filling error is given in section 2.5. Reco is not used further in the results section.*

Line 228. Sentence starting with 'We partitioned' belongs to the beginning of para 2.2.4.

*Thank you. We now corrected this (removal at line 233–234; addition at line 217–218).*

Line 310. Useful approach.

*Thank you.*

Line 413. This is a very relevant finding and should be presented also in the abstract.

*We now mention this in the abstract (line 27–28).*

Line 419 ff. Authors present results for alternative regression models but it is not explained what to do with this finding.

*The alternative regression models are mainly meant as an eye-opener to the community to generate awareness that the type of regression that is chosen can substantially affect the best estimate and confidence intervals of slopes and intercepts. Since there can be good arguments to choose any of the presented alternative regression models, we refrain from making any recommendations here. The simple linear regression receives most attention in our manuscript as it is compared to other relationships in literature that are based on simple linear regression.*

General: The MS would benefit from showing the year-round GW measurements in a supplementary figure (and not only aggregated as in Table S1) as this would visualize the effect of WIS.

*We added Figure S2 that contains timeseries graphs showing hourly values of the average water table depth (WTD) for the control (CON) and water infiltration system (WIS) plot for each of the studied locations.*

Points to be included in the discussion section:

Line 297. DOC and other pathways were not considered, which is acceptable given the efforts to estimate those terms. However, leaving them out from the overall budget calculation implies that the observed net C losses from the studied systems might be actually higher. This should be discussed later, together with literature estimates from similar systems (if available), denoting the possible magnitude of this effect.

*This is indeed an important topic, not only from the viewpoint of the magnitude of C emission related to this waterborne C export, but also considering a potential difference in waterborne C export between control and treatment sites that needs to be addressed. The newly added paragraph 3.5 (see our reply on page 1) discusses this matter in detail. Note that we did not include estimates from similar systems, since we are not aware of any studies of Dutch coastal peatlands that quantified waterborne C losses.*

Data in Fig. 6 show that manure affects the NECB. How much manure is produced from the harvests per site and how much would therefore be available to improve the NECB? This is more than a theoretical question as a the effect of manure on the ecosystem's C budget is now accounted for elsewhere.

*The main objective of this study was to quantify C budgets of the peat soil, rather than C budgets at the ecosystem scale (i.e., including farm-scale emissions). Hence, we did not investigate how much of the harvest at each site was fed to the cows and how much manure was produced from that. Regarding our studied plots, manure was only applied where necessary (i.e., the organic farm of location Aldeboarn [paragraph 2.4]).*

You discuss why are NEE/NECB results are so different to Tiemeyer et al. 2020. Could the clay layer also play a role? The SOC within the clay layer may be less prone to decomposition as unprotected peat, for example.

*The clay layer may indeed contribute to differences between our results and those of Tiemeyer et al. 2020. We now added a discussion on this matter to the main text (see our reply on page 2 for the contents of this discussion).*

Technical remarks

Line 68. Please update references Buzacott and van den Berg.

*We updated the references in question.*

Line 68. Sentence not complete. They mean a change from conventional agricultural land use towards what?

*We meant moving away from conventional agriculture to alternatives, such as wet agriculture (paludiculture) or rewetting for nature restoration. We now clarified this (line 72).*

Line 75. Please add: 'as the hydraulic conductivity of degraded peat soils is…'

*We now added 'degraded' to the sentence (line 80).*

Figure 2. I suggest to replace the grey bars for the WTD by blue bars or another colour providing better visibility. The figure provides C concentrations in kgC/m3; this is a C density, a C concentration would be kgC/kg soil.

*We now revised this figure and the description accordingly.*

Figure 4. Please revise so that either dark and light red is replaced by a different colour.

*The colors in the figure are on a continuous rather than a discrete scale. Hence, instead of replacing either dark or light red, we added a legend showing the number of half-hourly fluxes corresponding to a given shade of red.*

Line 328. Please move first parenthesis before '2022'.

*We now removed the parentheses.*

Figure 9b. I suggest to use also different symbol types, not only different colours in this plot.

*We now use different symbols in Figure 9b. We now also use additional line types in Figure 9a and a color blind-proof color palette in an attempt to make lines and symbols distinguishable for the various kinds of color blindness.*

Anonymous Referee #2, 5 Jun 2024

The manuscript is based on an impressive dataset and is clearly written and good English language. This kind of information on the effectiveness of GHG mitigation measures on peat soils and methods to estimate the emissions based on environmental variable is urgently needed. Comparison of the WTD-NECB relationship in different datasets (Fig. 9) was especially interesting in this manuscript. I have only minor comments.

*Thank you for the kind words and positive assessment of our work.*

Title: I'm not sure if the title should start with "Using automated transparent chambers to quantify…" as this was not a methodology-oriented paper to my opinion. I would stress the large dataset by including the number of sites and probably "continuous measurements" in the title if needed. Could it be: $CO_2$ emission reduction potential of water infiltration systems at six drained coastal peatland sites in the Netherlands.

*Thank you for the suggestion. We removed the automated transparent chamber part of the title and revised it to: "$CO_2$ emissions of drained coastal peatlands in the Netherlands and potential emission reduction by water infiltration systems"*

I think the infiltration system you used is also called submerged drainage. If this is true, please also include this term in the methods section to make it clearer for readers who are less familiar with (Dutch) drainage systems.

*This is true. We now added this clarification (line 103).*

In order to understand the functioning of the WIS, a figure on the WTD variation within a year would be useful.

*We now added Figure S2 that contains timeseries graphs showing hourly values of the average water table depth (WTD) for the control (CON) and water infiltration system (WIS) plot for each of the studied locations.*

Table 1: should the title in the 6th column be "ditch WTD", not "aim".

*We now revised this to "targeted ditch WT" to reflect that the values in this column are not measured values but ditch water tables targeted by the water authority.*

Line 347: is-->was

*We now corrected this (line 361).*

Line 370: add also the mean value (all sites) for NECB.

*We now added the mean value of the NECB across all sites (line 386).*

Line 412: This kind of observations on the proportion of C lost annually do not widely exist. You could add it to the abstract.

*We now added this to the abstract (line 27–28).*

Line 478: Paludiculture is not a water management system but a cultivation system. You could even use WIS to raise the WT for paludiculture (if possible to raise the WTD to 20 cm). I suggest revising this sentence e.g. to: Apart from WIS, typically leading to moderate WTD increase, more efficient WTD regulation could be implemented to allow paludiculture (Geurts et al., 2019; Martens et al., 2023) or restoration to a full peat growing ecosystem (Nugent et al., 2019).

*Thank you for the suggestion. We now revised the text accordingly (lines 501–506).*